# Complete Genome Sequences and Pathogenicity Analysis of Two Red Sea Bream Iridoviruses Isolated from Cultured Fish in Korea

**Min-A Jeong** [ID] **, Ye-Jin Jeong and Kwang-Il Kim** *[ID]

Department of Aquatic Life Medicine, Pukyong National University, Busan 48513, Korea;
jm0613@pukyong.ac.kr (M.-A.J.); 201513346@pukyong.ac.kr (Y.-J.J.)
* Correspondence: kimki@pknu.ac.kr; Tel.: +82-51-629-5946; Fax: +82-51-629-5938

**Abstract:** In Korea, red sea bream iridovirus (RSIV), especially subtype II, has been the main causative agent of red sea bream iridoviral disease since the 1990s. Herein, we report two Korean RSIV isolates with different subtypes based on the major capsid protein and adenosine triphosphatase genes: 17SbTy (RSIV mixed subtype I/II) from Japanese seabass (*Lateolabrax japonicus*) and 17RbGs (RSIV subtype II) from rock bream (*Oplegnathus fasciatus*). The complete genome sequences of 17SbTy and 17RbGs were 112,360 and 112,235 bp long, respectively (115 and 114 open reading frames [ORFs], respectively). Based on nucleotide sequence homology with sequences of representative RSIVs, 69 of 115 ORFs of 17SbTy were most closely related to subtype II (98.48–100% identity), and 46 were closely related to subtype I (98.77–100% identity). In comparison with RSIVs, 17SbTy and 17RbGs carried two insertion/deletion mutations (ORFs 014R and 102R on the basis of 17SbTy) in regions encoding functional proteins (a DNA-binding protein and a myristoylated membrane protein). Notably, survival rates differed significantly between 17SbTy-infected and 17RbGs-infected rock breams, indicating that the genomic characteristics and/or adaptations to their respective original hosts might influence pathogenicity. Thus, this study provides complete genome sequences and insights into the pathogenicity of two newly identified RSIV isolates classified as a mixed subtype I/II and subtype II.

**Keywords:** red sea bream iridoviral disease; red sea bream iridovirus; complete genome; insertion-deletion mutations; pathogenicity

## 1. Introduction

The virus species infectious spleen and kidney necrosis virus (ISKNV) (genus *Megalocytivirus*, family *Iridoviridae*) causes red sea bream iridoviral disease (RSIVD), which has a high mortality rate, in more than 30 susceptible freshwater and marine fish species [1]. According to the World Organization for Animal Health, it is a major fish disease [2]. Phylogenetic analyses based on major capsid protein (MCP) or adenosine triphosphatase (ATPase) genes have shown that the species can be classified into three major genotypes: red sea bream iridovirus (RSIV), ISKNV, and turbot reddish body iridovirus (TRBIV) [3]. The RSIV and ISKNV types can each be further categorized into two subtypes (I and II) [3]. Since the first outbreak of an RSIV-type infection among red sea breams (*Pagrus major*) in Japan in 1990 [4], RSIVs have been the predominant genotypes detected in marine fish in East Asian countries, including Korea [5–7]. In China, ISKNV and TRBIV types were first isolated from mandarin fish (*Siniperca chuatsi*) in 1998 [8,9] and from turbot (*Scophthalmus maximus*) in 2002, respectively [10]. In Korea, two genotypes of *Megalocytivirus* have been reported as endemic and have been taxonomically classified as RSIV [6,7,11] and TRBIV types [12]. Of note, RSIV subtype II has been identified as the major causative pathogen of endemic RSIVD in cultured marine fish in Korea [5].

Recently, an ISKNV/RSIV recombinant type was isolated from red sea bream (*Pagrus major*) in Taiwan, known as RSIV-Ku [13]. Its genome shares a high degree of homology with ISKNV-type viruses, except for specific nucleotide sequences that are closely related to RSIV-type viruses, implying that RSIV-Ku is a natural recombinant of ISKNV- and RSIV-type viruses [13]. Moreover, RSIV SB5-TY from a diseased Japanese seabass (*Lateolabrax japonicus*) in Korea is believed to be a genetic variant of RSIV-type viruses based on sequence difference in MCP and ankyrin repeat domains [5]. The emergence of genetic recombinants or variants of *Megalocytivirus* is a possibility, especially in RSIVD-endemic regions, such as Korea. Therefore, pathogenicity and complete genome sequence analyses of isolates in susceptible hosts are crucial for epidemiological studies, such as studies of source tracking and virus transmission.

In this study, we determined the complete genome sequences of two RSIVs identified in two cultured marine fish species (Japanese seabass and the rock bream (*Oplegnathus fasciatus*) in Korea, and analyzed insertion/deletion mutations (InDels). In addition, to evaluate their pathogenicity, a challenge test was performed on rock breams, which are known to be highly susceptible to RSIV infection.

## 2. Materials and Methods

### 2.1. Viral Culture

Primary cells derived from the fins of rock breams were grown in the L-15 medium supplemented with 10% fetal bovine serum (Performance Plus; Gibco, Grand Island, NY, USA) and 1% antibiotic-antimycotic solution (Gibco), as described by Lee et al. [14]. Briefly, caudal fin tissue was collected from juvenile rock bream (bodyweight, $5.4 \pm 0.8$ g), minced into small pieces (approximately 1 cm$^3$), and then washed with phosphate-buffered saline (PBS). Cells treated with a 0.25% trypsin-EDTA solution (Gibco) at 20 °C for 1 h were filtered through a cell strainer (pore size: 70 μm; Falcon, NY, USA). Filtered cells were collected via centrifugation at $500\times g$ for 10 min at 4 °C and were then resuspended in the culture medium and seeded in 25 cm$^2$ tissue culture flasks. The primary cells were incubated at 25 °C, and the medium was replaced daily. The cells were subcultured (split ratio: 1:2) when monolayer cells reached >90% confluence.

Tissue samples (spleen and kidney, 50 mg) were collected from diseased Japanese seabass in Tongyeong and rock bream in Goseong in 2017. To identify RSIV infection, real-time polymerase chain reaction (PCR) [15] was carried out. Briefly, each 20 μL real-time PCR mixture contained 1 μL of DNA, which was extracted using the yesG$^{TM}$ Cell Tissue Mini Kit (GensGen, Busan, Korea), 200 nM each primer and probe (Table A1), 10 μL of the $2\times$ HS Prime qPCR Premix (Genet Bio, Daejeon, Korea), 0.4 μL of the $50\times$ ROX dye, and 5.6 μL of nuclease-free water. Amplification was performed using a StepOne Real-time PCR system (Applied Biosystems, Foster City, CA, USA) under the following conditions: 95 °C for 10 min, followed by 40 cycles of 94 °C for 10 s (denaturation) and 60 °C for 35 s (annealing and extension). Tissue samples that were RSIV-positive, as determined by real-time PCR, were used as the viral inoculum.

Viral infection (each tissue homogenate, 10 mg/mL) was induced in 75 cm$^2$ tissue culture flasks (Greiner Bio-one, Frickenhausen, Germany) containing monolayers of primary cells at passage 15. RSIV-infected cells were propagated at 25 °C for 7 days in L-15 medium containing 5% fetal bovine serum and 1% antibiotic-antimycotic solution. After the appearance of the cytopathic effect (rounded cells; Figure A1), the infected cells were collected and subjected to three freeze-thaw cycles. After centrifugation at $500\times g$ for 10 min, the virus-containing supernatants were collected and stored at −80 °C until use. The cultured RSIVs were designated as 17SbTy and 17RbGs based on the sampling year, common name of the fish, and sampling site (i.e., 20<u>17</u>, Japanese <u>sea</u>b<u>a</u>ss, <u>T</u>ong<u>y</u>eong and 20<u>17</u>, <u>r</u>ock <u>b</u>ream, <u>Go</u>seong).

*2.2. Phylogenetic Analysis*

For genotyping, genes encoding MCP and ATPase were amplified with the primers listed in Table A1 and sequenced using an ABI 3730XL DNA Analyzer (Applied Biosystems, CA, USA) by Bionics Co. (Seoul, Korea). Then, the MCP and ATPase gene sequences were quality-checked by base-calling using ChromasPro (ver. 1.7.5; Technelysium, Tewantin, Australia). Each sequence was identified using Nucleotide Basic Local Alignment Search Tool (BLASTn; https://blast.ncbi.nlm.nih.gov/Blast.cgi). Contigs were generated using the ChromasPro and aligned using the ClustalW algorithm in BioEdit (ver. 7.2.5). Phylogenetic trees were generated by the maximum likelihood method via the Kimura two-parameter (K2P) model with a gamma-distribution and invariant sites (K2P + G4 + I) using MEGA (ver. 11). The MCP and ATPase genes of epizootic haematopoietic necrosis virus (GenBank accession no. FJ433873) were used as outgroup in the phylogenetic analyses. Support for specific genotypes of the RSIVs were determined with 1000 bootstrap replicates (≥70%).

*2.3. Determination of Complete Genome Sequences by Next-Generation Sequencing*

Viral nucleic acids were extracted from gradient-purified virions using the QIAamp MinElute Virus Spin Kit (Qiagen, Hilden, Germany). Next, 1 µg of the extracted DNA was employed to construct sequencing libraries using the QIAseq FX Single Cell DNA Library Kit (Qiagen). Sequencing libraries of 17SbTy and 17RbGs were constructed, with average lengths of 648 bp and 559 bp, respectively. The quality of the libraries was evaluated using the Agilent High Sensitivity D 5000 ScreenTape System (Agilent Scientific, CA, USA), and the quantity was determined using a Light Cycler Real-time PCR system (Roche, Mannheim, Germany). The high-quality libraries (300–600 bp) were sequenced (pair-end sequencing, 2 × 150 bp) by G&C Bio Co. (Daejeon, Korea) on the Illumina HiSeq platform (Illumina, San Diego, CA, USA). To assess the quality of the sequence data, FastQC [16] and MultiQC [17] were employed. Low-quality sequences (base quality <20) and the Illumina universal adapters were trimmed from the reads using Trim-Galore software (ver. 0.6.1; https://www.bioinformatics.babraham.ac.uk/projects/trim_galore, accessed on 21 June 2020). High-quality reads were mapped and assembled into contigs using gsMapper (ver. 2.8). Nucleotide errors in the reads were corrected with the Illumina sequencing data using Proovread [18].

*2.4. Complete Genome Sequence Analysis*

2.4.1. Construction of a Circular Map

The composition, structure, and homologous regions of the genomic DNA were analyzed and circular map was generated using the cgview comparison tool [19]. Coding regions were classified according to a clusters of orthologous groups (COG) analysis. To determine COG categories, a comparative analysis was performed based on the proteins encoded in 43 complete genomes representing 30 major phylogenetic lineages described by Tatusov et al. (1997 and 2001) [20,21] using the COG program on the National Center for Biotechnology Information (NCBI) website (http://www.ncbi.nlm.nih.gov/COG, accessed on 12 April 2021). The genes were categorized in accordance with their functional annotations.

2.4.2. Gene Annotation and Open Reading Frame (ORF) Analysis

To identify putative ORFs, the full-length genome sequences of 17SbTy and 17RbGs were annotated using Prokka (ver. 2.1). ORFs were predicted using NCBI ORFfinder (https://www.ncbi.nlm.nih.gov/orffinder, accessed on 15 April 2021), and then the amino acid sequences of the putative ORFs were checked by Protein BLAST (BLASTp; https://blast.ncbi.nlm.nih.gov/Blast.cgi, accessed on 16 April 2021). Nucleotide sequence homologies of the putative ORFs of 17SbTy with those of 17RbGs and representative megalocytiviruses, i.e., Ehime-1 (GenBank accession no. AB104413; RSIV subtype I and the ancestral strain of RSIVD) [22], ISKNV (GenBank accession no. AF371960) [8], and TRBIV (GenBank accession no. GQ273492)] [23] were determined using BLAST (https://blast.ncbi.nlm.nih.

gov/Blast.cgi, accessed on 10 May 2021). Furthermore, to analyze genetic relatedness among viruses in *Iridoviridae*, amino acid sequences of 26 conserved genes [24,25] were retrieved from NCBI GenBank. A phylogenetic tree based on the deduced amino acid sequences of 26 concatenated genes was constructed by the maximum likelihood method with the LG model and gamma-distributed rates with invariant sites (LG + G4 + I) [26] using MEGA (ver. 11.). Support for specific genera of iridoviruses was determined with 1000 bootstrap replicates ($\geq$70%).

### 2.4.3. Analysis of InDels in RSIVs

To identify InDels in coding regions, the nucleotide sequences of 17SbTy and 17RbGs were compared with those of the ancestral RSIV (Ehime-1 isolated from a red sea bream in Japan in 1990; RSIV subtype I) [22] and an RSIV genome previously reported in Korea (RBIV-KOR-TY1 isolate found in a rock bream in 2000; RSIV subtype II; GenBank accession no. AY532606) [27]. Genomic sequences coding for functional proteins were aligned using the ClustalW algorithm in BioEdit (ver. 7.2.5), and InDels in the coding regions were detected.

### 2.5. Pathogenicity of the Two RSIV Isolates in the Rock Bream

Healthy rock bream (body length: 8.75 $\pm$ 1.95 [mean $\pm$ SD]; body weight: 6.79 $\pm$ 4.16 g) were obtained from an aquaculture farm in Geoje, Korea, after confirming that they were RSIV-free by PCR, as described in the Manual of Diagnostic Tests for Aquatic Animals for RSIVD [2,28], and by real-time PCR [15] (Table A1). The fish were acclimated in a 500 L aqua tank at 25.0 $\pm$ 0.5 °C for 2 weeks and were fed a commercial diet once daily. Each day, 50% of rearing water was replaced with temperature-adjusted (25 °C) fresh seawater. To prepare a viral inoculum, viral genome copy numbers of cultured 17SbTy and 17RbGs were determined by real-time PCR [15] with a standard curve constructed using the serial dilutions of a plasmid containing the MCP gene of 17RbGs. In a challenge test, each fish group was intraperitoneally injected with 0.1 mL of 17SbTy ($n$ = 18; $10^4$ viral genome copies per fish), 17RbGs ($n$ = 18; $10^4$ viral genome copies per fish), or PBS ($n$ = 18; a negative control). After the viral challenge, the fish were maintained at 25.0 $\pm$ 0.5 °C in 30 L aqua tanks for 3 weeks, with 50% of water exchanged daily. DNA was extracted from the spleen tissue of dead fish, and RSIV infection was confirmed by real-time PCR. Survival rates were compared among the experimental groups by the log-rank test using GraphPad Prism (ver. 8.4.3.). Statistical significance was set at $p$-values < 0.05. Furthermore, the nucleotide sequences around four InDels in coding regions (ORFs 014R, 053R, 054R, and 102R on the basis of the 17SbTy isolate) were compared between cell-cultured isolates and viruses from RSIV-infected fish. DNA was extracted from three fish in each experimental group, and PCRs were carried out with each specific primer set (Table A1). Each 20 μL PCR mixture contained 1 μL of DNA (extracted using the yesG™ Cell Tissue Mini Kit; GensGen, Korea), 500 nM each primer, 10 μL of the 2× ExPrime Taq Premix (Genet Bio, Daejeon, Korea), and 7 μL of nuclease-free water. Amplification was performed on an Alpha Cycler 1 machine (PCRmax, Staffordshire, UK) under the following conditions: 95 °C for 10 min, followed by 35 cycles at 94 °C for 30 s (denaturation), 55 °C for 30 s (annealing), and 72 °C for 60 s (extension). The amplicons were sequenced using the ABI 3730XL DNA Analyzer (Applied Biosystems) by Bionics Co. Contigs were assembled using ChromasPro (ver. 1.7.5) and aligned using the ClustalW algorithm in BioEdit (ver. 7.2.5).

### 3. Results & Discussion

The complete genome sequences of two RSIV isolates collected from representative fish susceptible to RSIVD (17SbTy from a Japanese seabass and 17RbGs from a rock bream) in Korea were investigated, and a comparative analysis of the pathogenicity of the isolates was performed. A phylogeny based on genes encoding MCP and ATPase revealed that 17RbGs belongs to RSIV subtype II, which has been the predominant genotype in marine fish in Korea since the 1990s [5]. Notably, 17SbTy grouped with subtype I or II of RSIV in

phylogenetic analyses based on MCP or ATPase, respectively (Figure 1). Comparisons of 17SbTy with Ehime-1 (ancestral RSIV subtype I) and 17RbGs (RSIV subtype II), showed 99.63% and 98.24% identity for the *MCP* gene and 99.03% and 100% identity for the *ATPase* gene, respectively. Golden mandarin fish iridovirus, an RSIV subtype I reported in Korea in 2016 [29], shares 99.9% sequence homology with Ehime-1 in both the MCP and ATPase genes. Unlike golden mandarin fish iridovirus, 17SbTy was classified as a mixed RSIV subtype (subtype I/II).

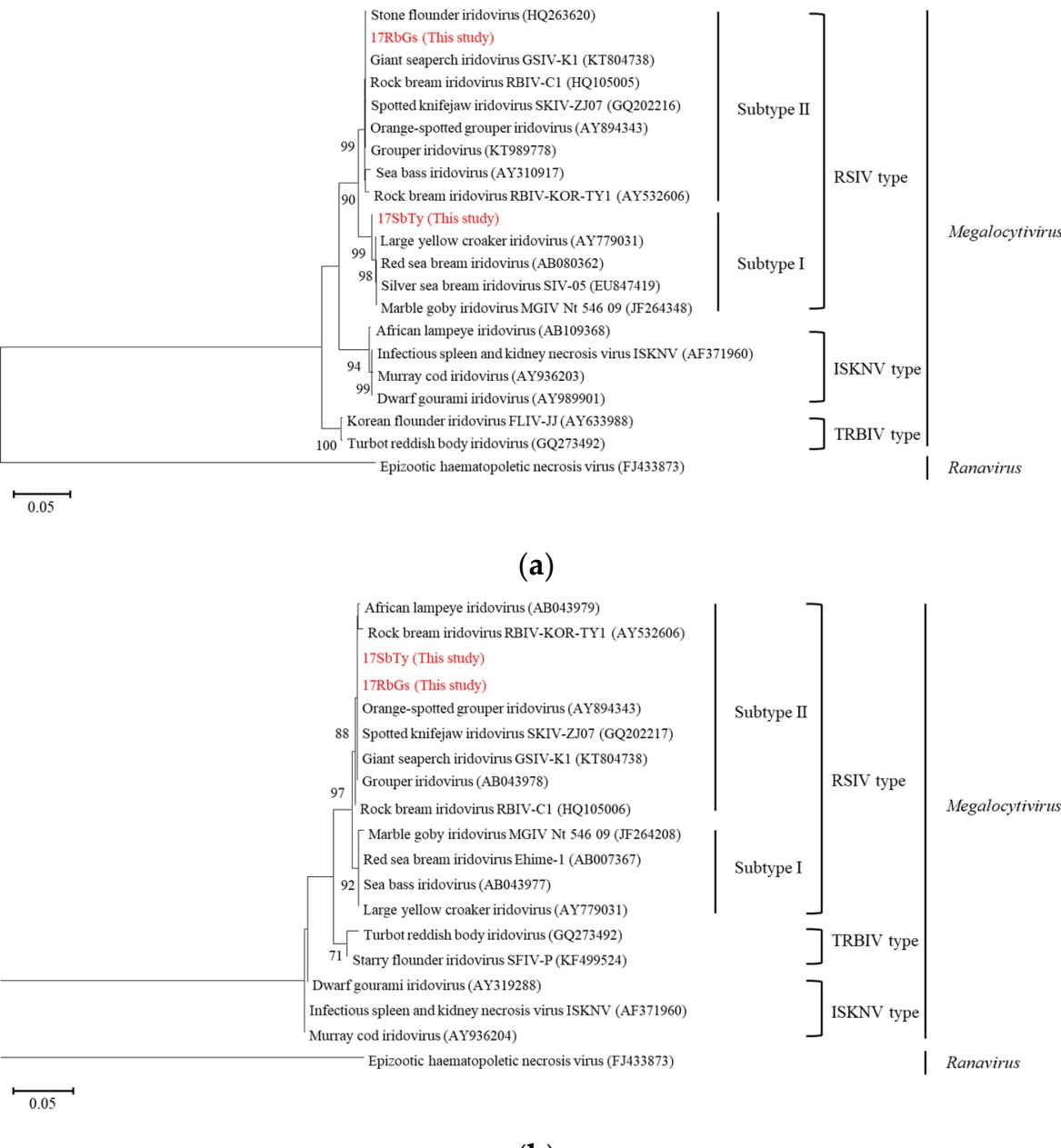

**Figure 1.** Phylogenetic trees based on the complete nucleotide sequences of the (**a**) major capsid protein gene (MCP; 1362 bp) and (**b**) adenosine triphosphatase gene (ATPase; 721 bp) of two red sea bream iridovirus (RSIV) isolates (17SbTy and 17RbGs) collected from cultured fish in Korea. The phylogenetic trees were constructed using the maximum-likelihood method in MEGA (ver. 11). Bootstrap values were obtained from 1000 replicates, and the scale bar represents 0.05 nucleotide substitutions per site. The two RSIV isolates (17SbTy and 17RbGs) from this study are highlighted in bold and red color.

The complete genomes of 17SbTy (122,360 bp, GenBank accession no. OK042108), and 17RbGs (122,235 bp, GenBank accession no. OK042109) were similar in size to the genomes of most representative megalocytiviruses, RSIV (Ehime-1; 112,415 bp), ISKNV (112,080 bp), and TRBIV (110,104 bp), except for scale drop disease virus (GF_MU1; GenBank accession no. MT521409; 131,129 bp). The sequences were circularly permuted and assembled into a circular form, similar to most *Megalocytivirus* genomes (Figure 2). In addition, the G+C contents of the 17SbTy and 17RbGs genomes were 53.28% and 53.13%, respectively.

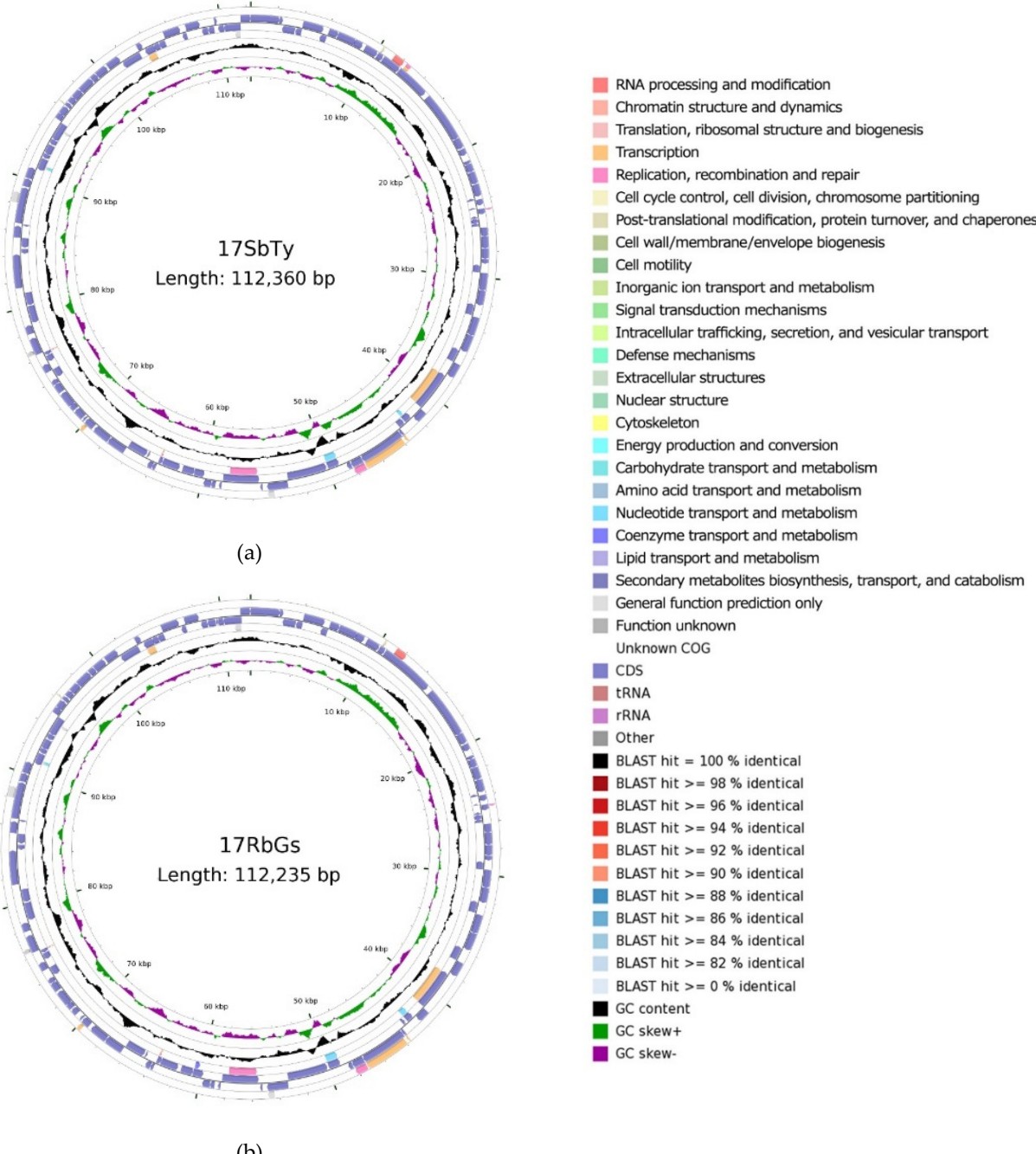

(a)

(b)

**Figure 2.** Circular genome maps of (**a**) 17SbTy (112,360 bp) and (**b**) 17RbGs (112,235 bp). From the inner ring to the outer ring, the first and eighth circles represented the genomic length (kbp) and nucleotide positions, respectively. The second and third circles show the G+C skew and G+C content, respectively. The fourth and fifth circles represent rRNA and tRNA genes on forward and reverse strands, respectively. The sixth and seventh circles indicate the functional categories of the protein-coding sequences in terms of clusters of orthologous groups (COG) on the forward and reverse strands, respectively.

In total, 115 and 114 putative ORFs were predicted in 17SbTy and 17RbGs, respectively (Table A2). The putative ORFs of 17SbTy (total length 104,868 bp, 93.3% of the genome) ranged in size from 111 to 3849 bp and encodes 36 to 1282 amino acid residues. Of the 115 ORFs, 70 were located on the sense (R) strand, and 45 were on the anti-sense (L) strand (Table A2). The putative ORFs of 17RbGs (total length 105,003 bp, 93.6% of genome) ranged in size from 111 to 4155 bp, encoding for 36 to 1384 amino acid residues. Of the 114 ORFs, 68 were located on the R strand and 46 were on the L strand. Of the annotated ORFs in 17SbTy (115 ORFs) and 17RbGs (114 ORFs), 43 (37.7%) and 42 (36.8%), respectively, could be assigned to a predicted structure and/or functional protein. The complete nucleotide sequences of 17SbTy and 17RbGs were closely related to rock bream iridovirus-C1 (RBIV-C1, GenBank accession no. KC244182) with identities of 99.56% and 99.69%, respectively. A comparison of the complete nucleotide sequences of 17SbTy and 17RbGs revealed 97.69% identity. In the ORFs of 17SbTy, nucleotide sequence identities were 87.99–100% with Ehime-1 (RSIV subtype I), 88.22–100% with 17RbGs (RSIV subtype II), 86.07–97.58% with ISKNV, and 80.25–99.66% with TRBIV (Table A2). Notably, the best matches for the nucleotide sequences of the 115 ORFs of 17SbTy were RSIV subtype II viruses (97.48–100% identity for 69 ORFs) and RSIV subtype I viruses (98.77–100% identity for 46 ORFs).

A total of 20 protein-coding genes in both 17SbTy (17.39%; 20/115 ORFs) and 17RbGs (17.54%; 20/114 ORFs) were annotated in the COG database, and these genes were assigned to nine functional groups (Table A3): (i) amino acid transport and metabolism; (ii) nucleotide transport and metabolism; (iii) translation, ribosomal structure, and biogenesis; (iv) transcription; (v) replication, recombination, and repair; (vi) signal transduction mechanisms; (vii) mobilome, prophages, transposons; (viii) general function prediction only; and (ix) function unknown. The nine functional groups identified in both 17SbTy and 17RbGs belonged to four major categories: metabolism, information storage and processing, cellular processes, and poorly characterized. Furthermore, both 17SbTy and 17RbGs harbored the 26 conserved genes that were shared by all members of the family *Iridoviridae*, including genes encoding enzymes and structural proteins involved in viral replication, transcriptional regulation, protein modification, and host-pathogen interactions [24,25]. The ORFs corresponding to these 26 core genes are listed in Table A4. A phylogenetic tree based on the concatenated amino acid sequences of the 26 conserved genes revealed that 17SbTy and 17RbGs can be assigned to the genus *Megalocytivirus*. Furthermore, 17SbTy was closely related to Ehime-1 (Figure 3).

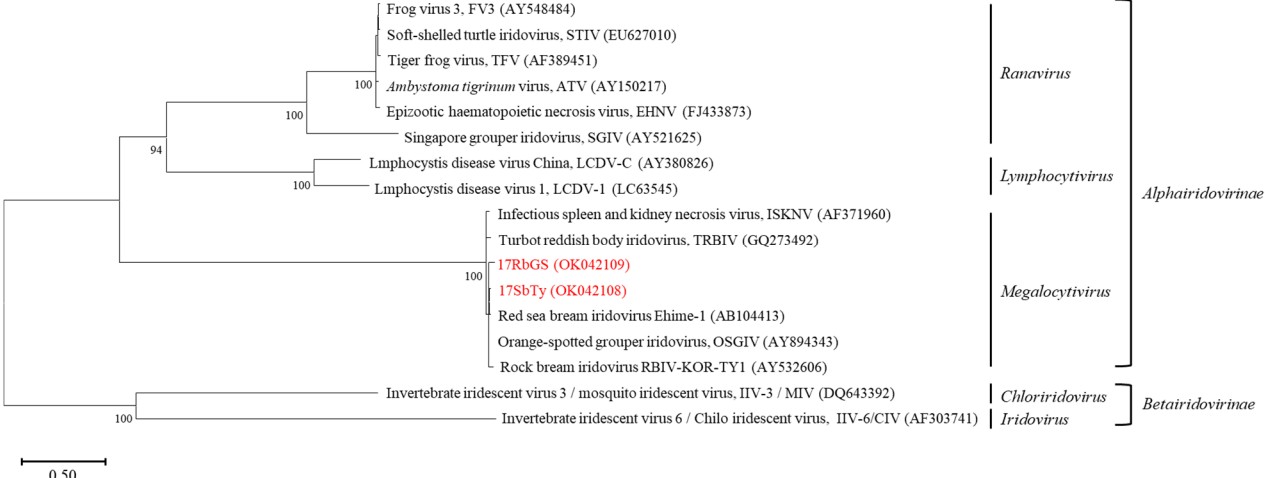

**Figure 3.** Phylogenetic trees based on the deduced amino acid sequences of the 26 concatenated genes conserved for members of the family *Iridoviridae*. The tree was constructed by the maximum-likelihood method under the LG model and gamma-distributed rates with invariant sites (LG + G4 + I) in MEGA (ver. 11). The two RSIV isolates (17SbTy and 17RbGs) from this study are highlighted in bold and red color.

As described by Eaton et al. (2007) [24], several annotated genes within the family *Iridoviridae* contain frameshift mutations. InDels are a type of frameshift mutation that can affect the translation of a functional protein. The complete genome of 17SbTy showed 133 InDels when compared to the Ehime-1 and 17RbGs genomes (data not shown). Notably, although the genomes of several RSIVs, including 17SbTy, Ehimel-1, and RBIV, encode two functional proteins—an mRNA-capping enzyme (ORF 012R, positions 10,693–12,165 in the 17SbTy genome) and a putative NTPase I (ORF 013R, positions 12,205–14,853 in the 17SbTy genome)—17RbGs possesses only a single functional protein (ORF 012R, positions 10,690–14,844 in the 17RbGs genome; Figure 4). A frameshift mutation caused by a short InDel [a 6 bp deletion, including a stop codon (TGA) and an intergenic codon (CCT)] explained the difference in the total number of ORFs between 17RbGs (*n* = 114) and 17SbTy (*n* = 115; Figure 4 and Table A2).

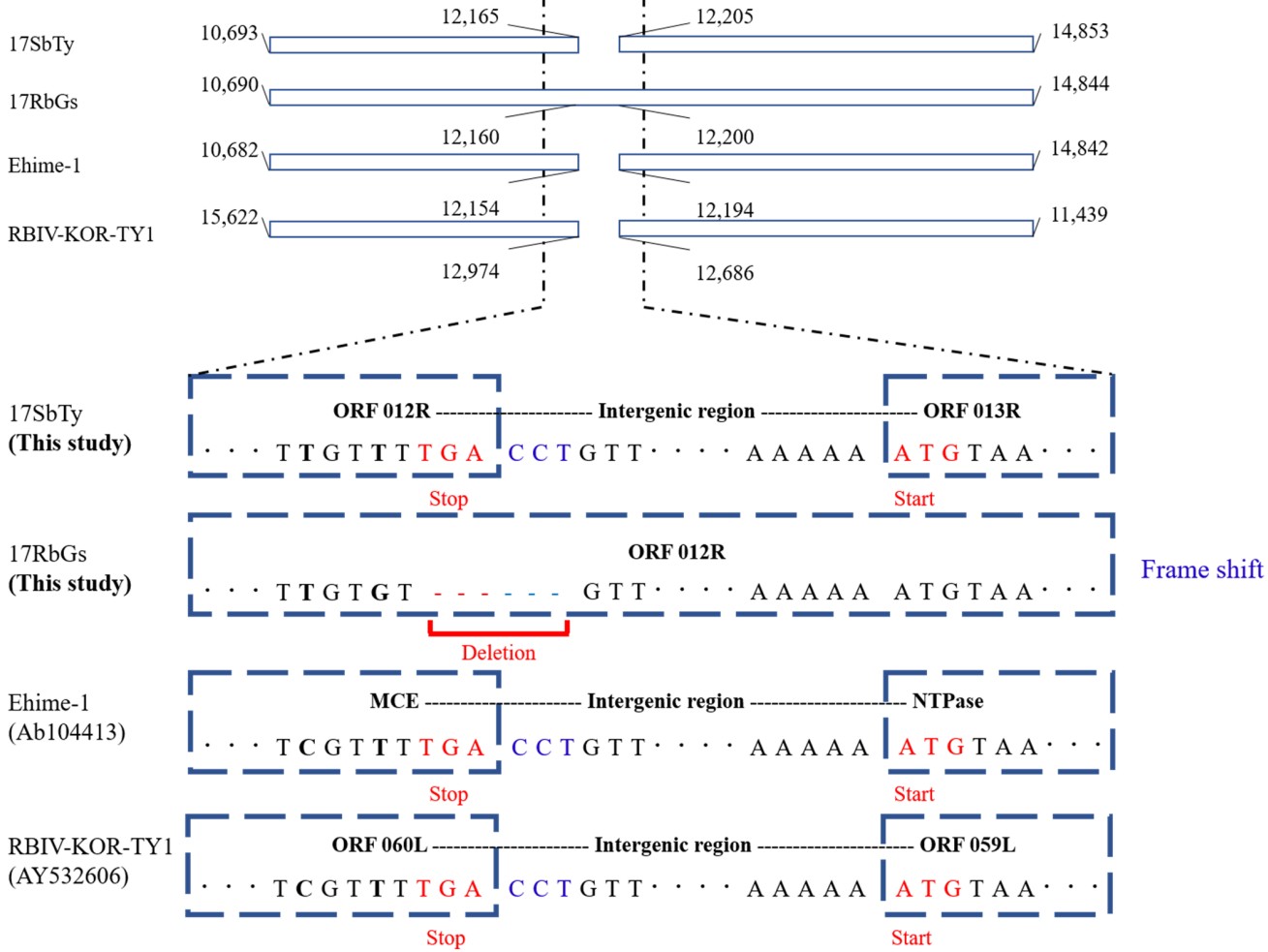

**Figure 4.** Schematic representation of a deletion of the termination codon in ORF 012R of 17RbGs causing a frameshift mutation. The aligned sequences are genomes of 17SbTy, 17RbGs, and two representative RSIVs (Ehime-1 [RSIV subtype I] and RBIV-KOR-TY1 [RSIV subtype II]). The nucleotide sequences surrounded by blue dashed lines are coding regions. The termination and start codons are shown in red, and the deleted sequences in the intergenic region are highlighted in blue.

Among the InDel regions in 17SbTy identified in comparisons with the Ehime-1 and 17RbGs genomes, 18 regions contained >10 bp mutations, and only four InDels were identified in coding regions (ORFs 014R, 053R, 054R, and 102R in 17SbTy). Although two ORFs encode known functional proteins (ORF 014R, which is involved in DNA binding, and ORF 102R, which is a myristoylated membrane protein; Figure 5a,d), two additional ORFs (ORF 053R and 054R) have not yet been functionally characterized (Figure 5b,c). Of

the InDels found in the ORFs known to encode functional proteins, a 27 bp deletion in a DNA-binding protein with an FtsK-like domain was identified in 17SbTy (ORF 014R), in 17RbGs (ORF 013R), and RBIV-KOR-TY 1 (ORF 058L), but not in Ehime-1 (ORF 077R; Figure 5a). The FtsK-like domain in spotted knifejaw iridovirus (an RSIV-type) [30] participates in host immune evasion by inhibiting transcriptional activities of NF-κB and INF-γ, indicating that the deleted sequences in the gene encoding a DNA-binding protein might affect viral replication and/or pathogenicity. Furthermore, ORF 102R of 17SbTy, located in the same region as ORF 575R in Ehime-1, encodes a myristoylated membrane protein, known as a viral envelope membrane protein of iridovirus, and its function may be conserved throughout the family *Iridoviridae* [31]. Thus, an InDel in the coding region of a viral membrane protein (a 30 bp deletion in ORF 101R of 17RbGs) may alter the regulation of viral entry into host cells at the onset of the infection cycle.

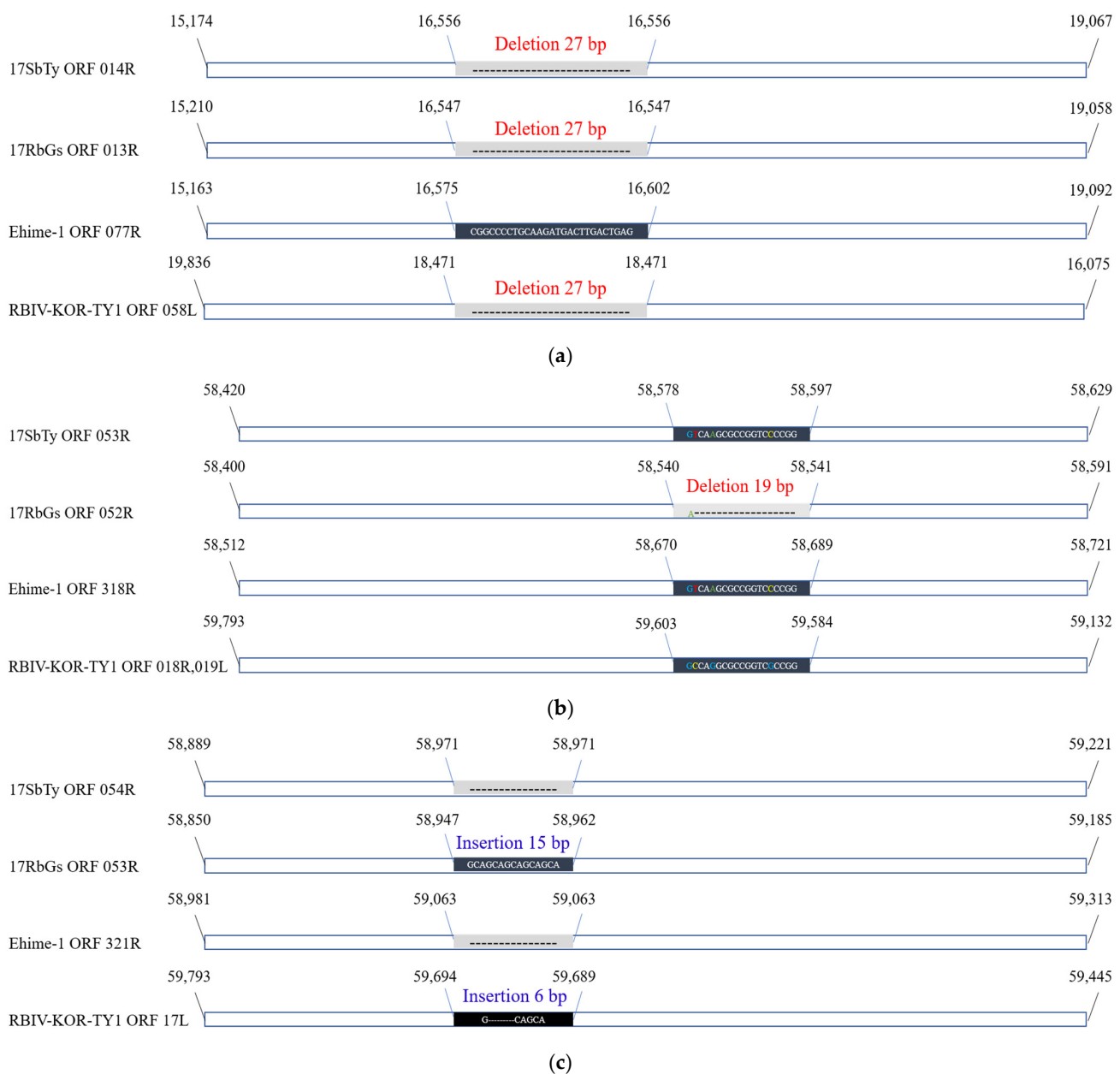

**Figure 5.** *Cont.*

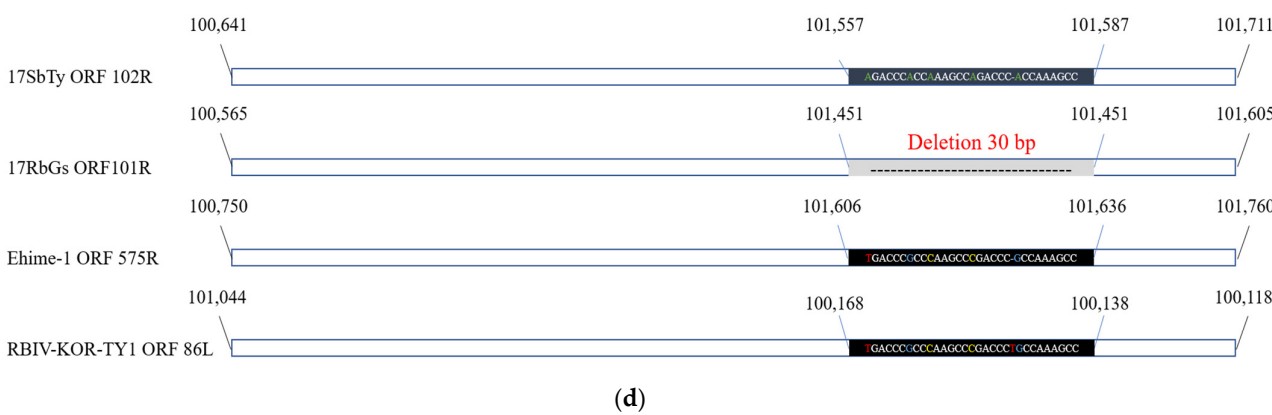

(d)

**Figure 5.** Schematic representation of insertion/deletion mutations (InDels) (>10 bp) in the coding regions as (**a**) ORF 014R, (**b**) ORF 053R, (**c**) ORF 054R and (**d**) ORF 102R based on the 17SbTy when compared with the genomes of 17RbGs and two representative RSIVs (Ehime-1 [RSIV subtype I] and RBIV-KOR-TY1 [RSIV subtype II]). Numbers indicate the positions of the InDels in the genome; white bars represent genome fragments, black bars denote insertions, and gray bars represent deletions.

No rock bream infected with 17RbGs survived 15 days post-injection, whereas 27.8% (5/18) of the 17SbTy-infected rock bream survived 21 days post-injection (Figure 6). The difference in survival rates between the 17SbTy- and 17RbGs-infected rock breams was significant (log-rank test, $p < 0.001$). The nucleotide sequences of the four InDel regions (ORFs 014R, 053R, 054R and 102R on the basis of the 17SbTy isolate) were identical in the cell-cultured isolates and viruses from dead fish (Figure A2). These results suggest that several of the genetic factors identified in the genomic analysis, including the InDels in coding regions, may influence virulence. Another noteworthy observation is that the apparent difference in virulence between the RSIV isolates may be due to adaptations to their respective original hosts (Japanese seabass for 17SbTy and rock bream for 17RbGs). Further molecular epidemiological studies, including analyses of RSIV replication and pathogenic determinants, are needed to elucidate the transmission of RSIV.

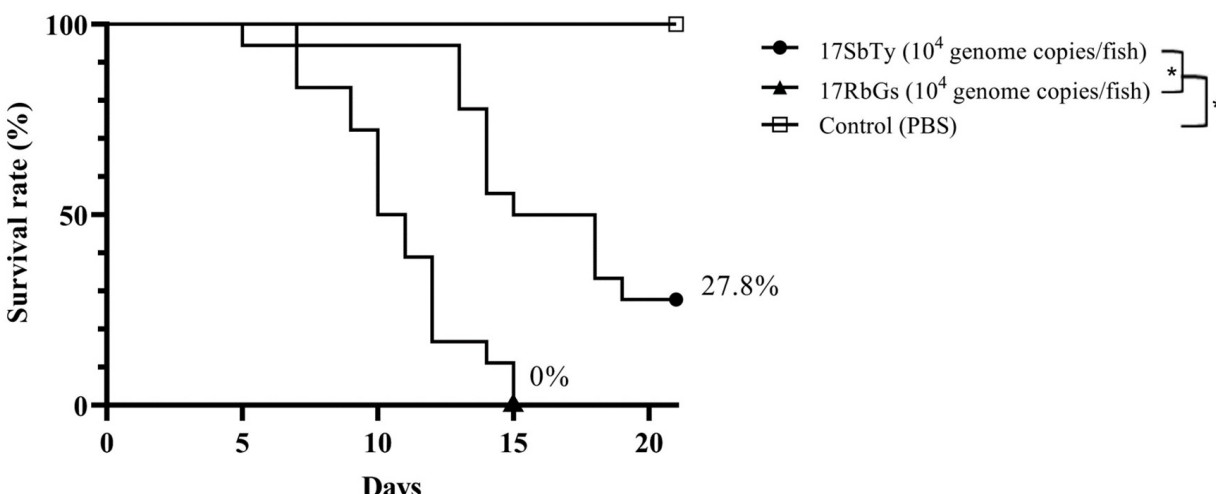

**Figure 6.** Survival rates (%) of rock breams after intraperitoneal injection with the two RSIV isolates (either 17SbTy or 17RbGs, $10^4$ genome copies per fish). Statistical analysis was performed by the log-rank test (* $p < 0.05$).

## 4. Conclusions

Phylogenetic trees based on genes encoding MCP and ATPase revealed that two RSIV isolates (17SbTy from a Japanese seabass and 17RbGs from a rock bream) can be

classified as RSIV mixed subtype I/II and subtype II, respectively. According to complete genome analysis, these isolates (17SbTy, 112,360 bp; 17RbGs, 112,360 bp) have the genomic organization, G+C content, coding capacity, and conserved core genes typical of the species *ISKNV*. Notably, the best matches for the nucleotide sequences in the 115 ORFs of 17SbTy were RSIV subtype II (69 matching ORFs; 97.48–100% identity) and RSIV subtype I (46 matching ORFs; 98.77–100% identity). In comparison with RSIVs, 17SbTy and 17RbGs had InDels in ORFs 014R and 102R (based on the 17SbTy genome), encoding a DNA-binding protein and myristoylated membrane protein, respectively. The survival rates of rock breams infected with these isolates differed significantly, suggesting that the genomic differences between these viruses and/or adaptations to their respective original hosts may have altered their pathogenicity. Thus, the complete genome sequences of these RSIV isolates provide basic information for molecular epidemiology and are expected to provide insight into viral replication in general and the pathogenicity of these viruses in susceptible hosts in particular.

**Author Contributions:** Conceptualization, M.-A.J. and K.-I.K.; methodology, M.-A.J. and Y.-J.J.; software, M.-A.J.; formal analysis, M.-A.J. and Y.-J.J.; writing—original draft preparation, M.-A.J.; writing—review and editing, K.-I.K.; project administration, K.-I.K.; funding acquisition, K.-I.K. All authors have read and agreed to the published version of the manuscript.

**Funding:** This research was funded by the National Research Foundation of Korea (NRF) from the Korean government (MSIT) grant number NRF-2021R1F1A1049419.

**Institutional Review Board Statement:** Animal experiment was performed with the approval of the Animal Ethics Committee of the Pukyong National University (Permission No. PKNUIACUC-2021-33).

**Informed Consent Statement:** Not applicable.

**Data Availability Statement:** Publicly available datasets were analyzed in this study. The full genome sequences generated in this study can be found in the National Center for Biotechnology Information (NCBI) GenBank (Accession No. OK042108 and OK042109).

**Acknowledgments:** We thank Hong-Seog Park (G&C Bio Co., Korea) for assistance with whole-genome sequencing.

**Conflicts of Interest:** The authors declare no conflict of interest.

## Appendix A

Table A1, PCR primers used in this study; Table A2, Predicted ORFs based on a comparison of isolates 17SbTy to 17RbGs and representative ISKNVs; Table A3, The coding sequences (CDSs) determined via COG classification of 17SbTy and 17RbGs in four functional categories; Table A4, ORF locations of the 26 conserved core genes conserved in the family *Iridoviridae*. Figure A1, Cytopathic effects (CPEs) in rock bream fin cells under the influence of a tissue homogenate from (A) an RSIV (17SbTy)-infected Japanese seabass and (B) an RSIV (17RbGs)-infected rock bream; Figure A2, Comparison of nucleotide sequences covering the four InDels in coding regions (ORFs 014R, 053R, 054R and 102R on the basis of the 17SbTy isolate) between the cell-cultured isolates and viruses from RSIV-infected rock breams.

**Table A1.** PCR primers used in this study.

| Primer | Target | Sequence(5′-3′) | Reference |
|--------|--------|-----------------|-----------|
| MCP 1F | | ATG TCT GCR ATC TCA GGT GC | |
| MCP 300R | | CCA GCG RAT GTA GCT GTT CTC | |
| MCP 600F | Major capsid protein | CAA GCT GCG GCG CTG GGA GG | [29] |
| MCP 800R | | GGC GCC ACC TGR CAC TGY TC | |
| MCP 1015F | | CTC ATT TTA CGA GAA CAC CC | |
| MCP 1362R | | TYA CAG GAT AGG GAA GCC TGC | |
| ATPase 1F | | ATG GAA ATC MAA GAR TTG TCC YTG | |
| ATPase 218R | ATPase | CAG TTR GGC AAY AGC TTG CT | This study |
| ATPase 529F | | GGG GGY AAC ATA CCM AAG C | |
| ATPase 721R | | CTT GCT TAC RCC ACG CCA G | |
| RSIV 1094F | | CCA GCA TGC CTG AGA TGG A | |
| RSIV 1221R | Major capsid protein | GTC CGA CAC CTT ACA TGA CAG G | [15] |
| RSIV 1177 probe | | FAM-TAC GGC CGC CTG TCC AAC G-BHQ1 | |
| 1-F | *Pst* I fragment | CTC AAA CAC TCT GGC TCA TC | |
| 1-R | | GCA CCA ACA CAT CTC CTA TC | [28] |
| 4-F | DNA polymerase gene | CGG GGG CAA TGA CGA CTA CA | |
| 4-R | | CCG CCT GTG CCT TTT CTG GA | |
| 14R-1F | | ATG AAG AAA TTT GAT TTT TGY RKA TGT C | |
| 14R-260R | | TCA TCC TCA GAG TCG CGG | |
| 14R-430F | | GCT CAG TTG TTC AAG ATG CC | |
| 14R-999R | | ATG CGT ATC ACA GTA CGC G | |
| 14R-848F | | CCA TAG AGG ATA ACA GCG C | |
| 14R-1202F | | ACA AGC GGG ACC TAT GCA A | |
| 14R-1841R | | TAC ATC GGC TCC TCA ACT G | |
| 14R-1620F | ORF 014R * | AGA ACT GGA GGA CTC ACA | |
| 14R-2011F | | CAC CGT GAA CTG CGC ATC T | This study |
| 14R-2630R | | GTC AGG TAT GTT TCC TGG TGT | |
| 14R-2309F | | GTA TGA TCG AGG AGA TCG CA | |
| 14R-2740F | | GAA CAC CGA GAG AGT GGA GAT G | |
| 14R-3241R | | AGT AGT CTA CCA CAG TTG C | |
| 14R-3190F | | TGT CAG CTA AAG GTC AGT GAT G | |
| 14R-3494F | | GTA TGT TGG ACT ACA TCG ACC C | |
| 14R-3849R | | TCA TTG ATT TTC ATT YAC ACC MAG | |
| 53R-1F | | ATG CCA CAG CCY ATT ATC TTC | |
| 53R-RB-192R | ORF 053R * | CTA AGC GCG CCT GGC TGG | |
| 53R-SB-210R | | CTA AGC AGC CCT GGC GGG | |
| ORF54-1F | ORF 054R * | ATG CCG ACT ACC AAA CAC A | |
| ORF54-348R | | TCA AAA CTC AAA GGC GCC G | |
| 102R-1F | | ATG AGT GCA ATA AAG GCA AAT G | |
| 102R-222R | | GTC CCG CAC GCC GTT GTT | |
| 102R-424F | ORF 102R * | CGC GTG CAT GCA ATG TAT | |
| 102R-797F | | GCA ATG TCT GTC AGG TGG C | |
| 102R-1071R | | CTA GGC AAA TGC AGC AAT AAC | |

* Open reading frame on the basis of 17SbTy isolate.

**Table A2.** Predicted ORFs based on a comparison of isolates 17SbTy to 17RbGs and representative ISKNVs.

| Gene ID 17SbTy | Position | | CDS Size (NT) | Predicted Structure and/or Function | Best-Match Homolog | | | Homolog to 17RbGs | | Homolog to Ehime_1 (AB104413.1) | | Homolog to ISKNV (AF371960) | | Homolog to TRBIV (GQ273492) | |
|---|---|---|---|---|---|---|---|---|---|---|---|---|---|---|---|
| | Start | End | | | Genotype | Isolates | Identity (%) | ORF no. | Identity (%) | ORF no. | Identity (%) | ORF no. | Identity (%) | ORF no. | Identity (%) |
| ORF 001R | 111,584 | 2196 | 2973 | hypothetical protein | RSIV subtype II | RSIV KagYT-96 RSIV RIE12-1 GSIV-K1 OSGIV | 99.70% | ORF 001R | 99.70% | ORF 639R | 98.18% | 76L | 93.44% | 69L | 92.91% |
| ORF 002R | 2198 | 2467 | 270 | hypothetical protein | RSIV subtype I | PIV2016 PIV2014a PIV2010 LYCIV RSIV Ehime-1 | 100.00% | ORF 002R | 96.67% | ORF 010R | 100.00% | 75L | 91.30% | 68L | 87.26% |
| ORF 003L | 2476 | 3495 | 1020 | hypothetical protein | RSIV subtype I | PIV2016 PIV2014a PIV2010 LYCIV RSIV Ehime-1 | 100.00% | ORF 003L | 98.53% | ORF 016L | 100.00% | 74R | 93.63% | 67R | 93.94% |
| ORF 004L | 3544 | 4032 | 489 | hypothetical protein | RSIV subtype I | PIV2016 PIV2014a PIV2010 LYCIV Zhoushan RSIV Ehime-1 | 100.00% | ORF 004L | 95.09% | ORF 019L | 100.00% | 73R | 90.24% | 66R | 84.72% |
| ORF 005R | 4015 | 5625 | 1611 | hypothetical protein | RSIV subtype I | LYCIV PIV2014a PIV2010 LYCIV Zhoushan RSIV Ehime-1 | 100.00% | ORF 005R | 98.08% | ORF 018R | 100.00% | 71L | 93.61% | 65L | 93.42% |
| ORF 006L | 5528 | 6043 | 516 | hypothetical protein | RSIV subtype I | PIV2014a PIV2010 LYCIV Zhoushan RSIV Ehime-1 | 100.00% | ORF 006L | 97.29% | ORF 026R | 100.00% | 70L | 95.20% | - | - |
| ORF 007R | 6065 | 6796 | 732 | hypothetical protein | RSIV subtype I | PIV2016 PIV2014a PIV2010 LYCIV Zhoushan RSIV Ehime-1 | 100.00% | ORF 007R | 96.86% | ORF 029R | 100.00% | 69L | 86.07% | 64L | - |
| ORF 008R | 6808 | 8241 | 1434 | hypothetical protein | RSIV subtype I | PIV2016 PIV2014a PIV2010 LYCIV Zhoushan RSIV Ehime-1 | 100.00% | ORF 008R | 97.63% | ORF 033R | 100.00% | 68L | 93.58% | 63L | 88.95% |
| ORF 009R | 8192 | 8860 | 669 | hypothetical protein | RSIV subtype I | LYCIV Zhoushan | 100.00% | ORF 009R | 98.06% | ORF 037R | 98.80% | 67L | 90.69% | 62L | 91.68% |
| ORF 010R | 9087 | 10,130 | 1044 | hypothetical protein | RSIV subtype II / ISKNV subtype I | RSIV KagYT-96 RSIV RIE12-1 GSIV-K1 RSIV-Ku LYCIV Zhoushan OSGIV | 100.00% | ORF 010R | 99.81% | ORF 042R | 98.46% | 66L | 92.82% | 61L | 92.53% |
| ORF 011R | 10,181 | 10,651 | 471 | RING-finger-containing E3 ubiquitin ligase | RSIV subtype II | RSIV KagYT-96 RSIV RIE12-1 RBIV-C1 LYCIV Zhoushan RSIV_121 17RbGs | 100.00% | ORF 011R | 100.00% | ORF 049R | 98.51% | 65L | 91.30% | 60L | 89.17% |

Table A2. *Cont.*

| Gene ID 17SbTy | Position | | CDS Size (NT) | Predicted Structure and/or Function | Best-Match Homolog | | | Homolog to 17RbGs | | Homolog to Ehime_1 (AB104413.1) | | Homolog to ISKNV (AF371960) | | Homolog to TRBIV (GQ273492) | |
|---|---|---|---|---|---|---|---|---|---|---|---|---|---|---|---|
| | Start | End | | | Genotype | Isolates | Identity (%) | ORF no. | Identity (%) | ORF no. | Identity (%) | ORF no. | Identity (%) | ORF no. | Identity (%) |
| ORF 012R | 10,693 | 12,165 | 1473 | mRNA capping enzyme | RSIV subtype II / ISKNV subtype I | RSIV KagYT-96 RSIV RIE12-1 GSIV-K1 RSIV-Ku LYCIV Zhoushan OSGIV | 100.00% | ORF 012R | 99.93% | MCE | 97.49% | 64L | 93.36% | 59L | 93.28% |
| ORF 013R | 12,205 | 14,853 | 2649 | putative NTPase I | RSIV subtype II | RSIV KagYT-96 RSIV RIE12-1 GSIV-K1 | 99.96% | - | - | NTPase | 97.92% | 63L | 93.36% | 58L | 93.42% |
| ORF 014R | 15,174 | 19,067 | 3849 | DNA-binding protein | RSIV subtype II | RSIV KagYT-96 RSIV RIE12-1 | 100.00% | ORF 013R | 99.48% | ORF 077R | 96.78% | 62L | 91.81% | 57L | 93.08% |
| ORF 015R | 19,064 | 19,870 | 807 | putative replication factor and/or DNA binding-packing | RSIV subtype II | RSIV KagYT-96 RSIV RIE12-1 GSIV-K1 RBIV-C1 RSIV_121 OSGIV | 100.00% | ORF 014R | 92.94% | ORF 092R | 97.65% | 61L | 93.80% | 56L | 93.06% |
| ORF 016R | 19,934 | 20,446 | 513 | hypothetical protein | RSIV subtype II | RSIV KagYT-96 GSIV-K1 OSGIV | 100.00% | ORF 015R | 89.35% | ORF 097R | 96.30% | 59L | 92.84% | 55L | 88.95% |
| ORF 017R | 20,918 | 21,178 | 261 | hypothetical protein | RSIV subtype II | RSIV KagYT-96 RSIV RIE12-1 SKIV RBIV-C1 RSIV_121 RBIV-KOR-TY1 OSGIV | 100.00% | ORF 016R | 95.40% | ORF 099R | 98.08% | 57L | 96.17% | 54L | 95.40% |
| ORF 018R | 21,185 | 21,832 | 648 | helicase family | RSIV subtype II | RSIV KagYT-96 GSIV-K1 OSGIV | 100.00% | ORF 017R | 99.23% | ORF 101R | 99.23% | 56L | 97.22% | 53L | 97.38% |
| ORF 019R | 21,843 | 22,784 | 942 | Serine-threonine protein kina | RSIV subtype II | RSIV KagYT-96 RSIV RIE12-1 GSIV-K1 SKIV RBIV-C1 RSIV_121 OSGIV 17RbGs | 100.00% | ORF 018R | 100.00% | ORF 106R | 96.92% | 55L | 90.98% | 52L | 89.81% |
| ORF 020R | 22,807 | 23,751 | 945 | hypothetical protein | RSIV subtype II | RSIV KagYT-96 GSIV-K1 SKIV RBIV-C1 RSIV_121 OSGIV 17RbGs | 100.00% | ORF 019R | 100.00% | ORF 111R | 97.67% | 54L | 90.08% | 51L | 90.48% |

**Table A2.** *Cont.*

| Gene ID 17SbTy | Position | | CDS Size (NT) | Predicted Structure and/or Function | Best-Match Homolog | | Identity (%) | Homolog to 17RbGs | | Homolog to Ehime_1 (AB104413.1) | | Homolog to ISKNV (AF371960) | | Homolog to TRBIV (GQ273492) | |
|---|---|---|---|---|---|---|---|---|---|---|---|---|---|---|---|
| | Start | End | | | Genotype | Isolates | | ORF no. | Identity (%) | ORF no. | Identity (%) | ORF no. | Identity (%) | ORF no. | Identity (%) |
| ORF 021L | 23,785 | 23,979 | 195 | hypothetical protein | RSIV subtype II | RSIV KagYT-96RSIV RIE12-1GSIV-K1SKIVRBIV-C1RSIV_121OSGIV17RbGs | 100.00% | ORF 020L | 100.00% | ORF 121L | 96.91% | 53R | 91.24% | 50R | - |
| ORF 022R | 23,981 | 24,433 | 453 | hypothetical protein | RSIV subtype II | RSIV KagYT-96 RSIV RIE12-1 GSIV-K1 SKIV RBIV-C1 RSIV_121 OSGIV 17RbGs | 100.00% | ORF 021R | 100.00% | ORF 122R | 96.47% | 52L | 88.91% | 49L | 88.21% |
| ORF 023L | 24,522 | 24,657 | 111 | hypothetical protein | RSIV subtype II | RSIV KagYT-96 RSIV RIE12-1 GSIV-K1 SKIV RBIV-C1 RSIV_121 RBIV-KOR-TY1 OSGIV 17RbGs | 100.00% | ORF 022L | 100.00% | ORF 127L | 93.86% | 51R | 91.46% | - | - |
| ORF 024R | 24,712 | 25,140 | 429 | hypothetical protein | RSIV subtype II | RSIV KagYT-96 RSIV RIE12-1 GSIV-K1 RBIV-KOR-TY1 OSGIV | 100.00% | ORF 023R | 99.77% | ORF 128R | 98.37% | 50L | 93.24% | 48L | 91.61% |
| ORF 025L | 25,208 | 25,378 | 171 | hypothetical protein | RSIV subtype II | RSIV KagYT-96 RSIV RIE12-1 GSIV-K1 SKIV RBIV-C1 RSIV_121 RBIV-KOR-TY1 OSGIV 17RbGs | 100.00% | ORF 024L | 100.00% | ORF 134L | 97.66% | 49R | 94.74% | - | - |
| ORF 026L | 25,394 | 25,747 | 354 | PDGF/VEGF-like protein ORF 135L | RSIV subtype II | RSIV KagYT-96 RSIV RIE12-1 GSIV-K1 OSGIV | 100.00% | ORF 025L | 99.72% | ORF 135L | 97.74% | 48R | 86.16% | 47R | 87.39% |
| ORF 027L | 25,744 | 26,007 | 264 | hypothetical protein | RSIV subtype II | RSIV KagYT-96 RSIV RIE12-1 GSIV-K1 SKIV RBIV-C1 RSIV_121 RBIV-KOR-TY1 OSGIV 17RbGs | 100.00% | ORF 026L | 100.00% | ORF 138L | 97.35% | 47R | 93.18% | 46R | 93.56% |

Table A2. *Cont.*

| Gene ID 17SbTy | Position | | CDS Size (NT) | Predicted Structure and/or Function | Best-Match Homolog | | | Homolog to 17RbGs | | Homolog to Ehime_1 (AB104413.1) | | Homolog to ISKNV (AF371960) | | Homolog to TRBIV (GQ273492) | |
|---|---|---|---|---|---|---|---|---|---|---|---|---|---|---|---|
| | Start | End | | | Genotype | Isolates | Identity (%) | ORF no. | Identity (%) | ORF no. | Identity (%) | ORF no. | Identity (%) | ORF no. | Identity (%) |
| ORF 028R | 26,167 | 26,850 | 684 | cytosine DNA methyltrans-ferase | RSIV subtype I | PIV2014a PIV2010 LYCIV Zhoushan RSIV Ehime-1 | 99.85% | ORF 027R | 97.95% | ORF 140R | 99.85% | 46L | 94.74% | 45L | 94.88% |
| ORF 029R | 26,844 | 27,758 | 915 | hypothetical protein | RSIV subtype I | PIV2016 PIV2014a PIV2010 LYCIV Zhoushan RSIV Ehime-1 | 100.00% | ORF 028R | 96.17% | ORF 145R | 100.00% | 45L | 88.74% | 44L | 89.84% |
| ORF 030R | 27,763 | 28,563 | 801 | hypothetical protein | RSIV subtype I | LYCIV Zhoushan RSIV Ehime-1 | 100.00% | ORF 029R | 97.50% | ORF 151R | 100.00% | 44L | 90.02% | 43L | 89.51% |
| ORF 031R | 28,570 | 28,932 | 363 | Erv1/Alr family | RSIV subtype I | PIV2016 PIV2014a PIV2010 LYCIV Zhoushan RSIV Ehime-1 | 100.00% | ORF 030R | 97.80% | ORF 156R | 100.00% | 43L | 94.21% | 42L | 95.04% |
| ORF 032L | 29,016 | 29,615 | 600 | hypothetical protein | RSIV subtype I | PIV2010 LYCIV Zhoushan RSIV Ehime-1 | 100.00% | ORF 031L | 96.83% | ORF 161L | 100.00% | 42R | 89.33% | 41R | 91.01% |
| ORF 033R | 29,630 | 30,979 | 1350 | hypothetical protein | RSIV subtype I | LYCIV Zhoushan | 100.00% | ORF 032R | 97.04% | ORF 162R | 99.56% | 41L | 88.96% | 40L | 90.53% |
| ORF 034R | 30,981 | 32,129 | 1149 | hypothetical protein | RSIV subtype I | LYCIV Zhoushan | 100.00% | ORF 033R | 91.91% | ORF 171R | 91.22% | 40L | 89.65% | 39L | 98.43% |
| ORF 035L | 32,122 | 33,000 | 879 | hypothetical protein | RSIV subtype I | LYCIV Zhoushan RSIV Ehime-1 LYCIV | 100.00% | ORF 034L | 93.97% | ORF 179L | 100.00% | 39R | 90.22% | 38R | 90.90% |
| ORF 036R | 33,066 | 34,505 | 1440 | hypothetical protein | RSIV subtype I | PIV2016 PIV2014a PIV2010 RSIV Ehime-1 | 100.00% | ORF 035R | 93.75% | ORF 180R | 100.00% | 38L | 90.71% | 37L | 90.90% |
| ORF 037R | 34,514 | 35,863 | 1350 | hypothetical protein | RSIV subtype I | PIV2016 PIV2014a PIV2010 LYCIV Zhoushan RSIV Ehime-1 | 99.93% | ORF 036R | 93.85% | ORF 186R | 99.93% | 37L | 90.11% | 36L | 90.96% |
| ORF 038L | 35,860 | 36,915 | 1056 | hypothetical protein | RSIV subtype I | PIV2010 LYCIV Zhoushan RSIV Ehime-1 LYCIV | 100.00% | ORF 037L | 95.17% | ORF 197L | 100.00% | 36R | 91.49% | 35R | 88.93% |
| ORF 039R | 36,909 | 38,048 | 1140 | hypothetical protein | RSIV subtype I | PIV2010 LYCIV Zhoushan | 100.00% | ORF 038R | 95.53% | ORF 198R | 99.91% | 35L | 88.64% | 34L | 88.88% |

Table A2. *Cont.*

| Gene ID 17SbTy | Position | | CDS Size (NT) | Predicted Structure and/or Function | Best-Match Homolog | | | Homolog to 17RbGs | | Homolog to Ehime_1 (AB104413.1) | | Homolog to ISKNV (AF371960) | | Homolog to TRBIV (GQ273492) | |
|---|---|---|---|---|---|---|---|---|---|---|---|---|---|---|---|
| | Start | End | | | Genotype | Isolates | Identity (%) | ORF no. | Identity (%) | ORF no. | Identity (%) | ORF no. | Identity (%) | ORF no. | Identity (%) |
| ORF 040L | 38,121 | 41,279 | 3159 | DNA dependent RNA polymerase second largest subunit | RSIV subtype I | LYCIV Zhoushan | 100.00% | ORF 039L | 96.52% | RPO-2 | 98.54% | 34R | 93.78% | 33R | 94.98% |
| ORF 041R | 41,362 | 42,264 | 903 | hypothetical protein | RSIV subtype I | LYCIV Zhoushan | 100.00% | ORF 040R | 95.90% | ORF 226R | 97.79% | 33L | 91.36% | 32L | 92.59% |
| ORF 042L | 42,327 | 42,943 | 582 | deoxyribo-nucleoside kinase | RSIV subtype I | LYCIV Zhoushan | 100.00% | ORF 041L | 88.87% | TK | 87.99% | 32R | 92.16% | 31R | 99.66% |
| ORF 043L | 43,008 | 43,535 | 243 | hypothetical protein | RSIV subtype I | PIV2016 PIV2014a PIV2010 RSIV Ehime-1 | 98.77% | ORF 042L | 95.47% | ORF 237L | 98.77% | 31.5L | 88.89% | 30R | 93.42% |
| ORF 044R | 43,603 | 43,824 | 222 | transcription elongation factor TFIIS | RSIV subtype I | PIV2016PIV2014aPIV2010LYCIV ZhoushanRSIV Ehime-1 | 100.00% | ORF 043R | 98.20% | ORF 238R | 100.00% | 29L | 96.40% | 29L | 97.06% |
| ORF 045R | 43,831 | 47,337 | 3507 | DNA dependent RNA polymerase largest subunit | RSIV subtype I | LYCIV Zhoushan PIV2016 PIV2014a PIV2010 | 99.94% | ORF 044R | 97.69% | RPO-1 | 99.37% | 28L | 94.66% | 28L | 95.30% |
| ORF 046R | 47,354 | 48,250 | 897 | probable XPG/RAD2 type nuclease | RSIV subtype I | PIV2016 PIV2014a PIV2010 LYCIV Zhoushan RSIV Ehime-1 | 100.00% | ORF 045R | 98.33% | ORF 256R | 100.00% | 27L | 96.10% | 27L | 95.21% |
| ORF 047R | 48,272 | 48,595 | 324 | hypothetical protein | RSIV subtype I | PIV2016 PIV2014a PIV2010 LYCIV Zhoushan RSIV Ehime-1 | 100.00% | ORF 046R | 97.53% | ORF 261R | 100.00% | 26L | 92.00% | 26L | 90.43% |

**Table A2.** *Cont.*

| Gene ID 17SbTy | Position | | CDS Size (NT) | Predicted Structure and/or Function | Best-Match Homolog | | | Homolog to 17RbGs | | Homolog to Ehime_1 (AB104413.1) | | Homolog to ISKNV (AF371960) | | Homolog to TRBIV (GQ273492) | |
|---|---|---|---|---|---|---|---|---|---|---|---|---|---|---|---|
| | Start | End | | | Genotype | Isolates | Identity (%) | ORF no. | Identity (%) | ORF no. | Identity (%) | ORF no. | Identity (%) | ORF no. | Identity (%) |
| ORF 048L | 49,064 | 50,002 | 939 | ribonucleotide diphosphate reductase small subunit | RSIV subtype I | PIV2016 PIV2014a PIV2010 RSIV Ehime-1 | 100.00% | ORF 047L | 98.08% | RR-2 | 100.00% | 24R | 94.68% | 25R | 95.21% |
| ORF 049L | 50,114 | 53,266 | 3153 | laminin-type epidermal growth factor | RSIV subtype I | PIV2010 RSIV Ehime-1 | 100.00% | ORF 048L | 93.77% | ORF 291L | 100.00% | 23R | 87.35% | 24R | 88.96% |
| ORF 050R | 53,339 | 54,934 | 1596 | LRP16 like protein macro domain-containing protein | RSIV subtype I | PIV2016 PIV2014a PIV2010 RSIV Ehime-1 | 100.00% | ORF 049R | 95.60% | ORF 292R | 100.00% | 22L | 93.41% | 23L | 93.52% |
| ORF 051R | 55,282 | 55,464 | 183 | hypothetical protein | RSIV subtype I | PIV2016 PIV2014a PIV2010 LYCIV Zhoushan RSIV Ehime-1 LYCIV | 100.00% | ORF 050R | 97.27% | ORF 300R | 100.00% | 20L | 89.95% | 21L | 94.54% |
| ORF 052L | 55,511 | 58,354 | 2844 | DNA polymerase family B exonuclease | RSIV subtype I | PIV2010 LYCIV Zhoushan RSIV Ehime-1 | 100.00% | ORF 051L | 97.23% | DPO | 100.00% | 19R | 95.11% | 20R | 93.15% |
| ORF 053R | 58,420 | 58,629 | 210 | hypothetical protein | RSIV subtype I | PIV2010 LYCIV Zhoushan RSIV Ehime-1 | 100.00% | ORF 052R | 92.55% | ORF 318R | 100.00% | 18.5L | 89.89% | 19L | 91.76% |
| ORF 054R | 58,889 | 59,221 | 333 | hypothetical protein | RSIV subtype I | PIV2016 PIV2014a PIV2010 LYCIV Zhoushan RSIV Ehime-1 | 100.00% | ORF 053R | 88.22% | ORF 321R | 100.00% | 17L | 92.81% | 17L | 89.47% |
| ORF 055R | 59,236 | 59,823 | 588 | hypothetical protein | RSIV subtype I | PIV2016 PIV2014a PIV2010 LYCIV Zhoushan RSIV Ehime-1 | 99.66% | ORF 054R | 92.35% | ORF 324R | 99.66% | 16L | 91.50% | 16L | 92.35% |

Table A2. *Cont.*

| Gene ID 17SbTy | Position | | CDS Size (NT) | Predicted Structure and/or Function | Best-Match Homolog | | | Homolog to 17RbGs | | Homolog to Ehime_1 (AB104413.1) | | Homolog to ISKNV (AF371960) | | Homolog to TRBIV (GQ273492) | |
|---|---|---|---|---|---|---|---|---|---|---|---|---|---|---|---|
| | Start | End | | | Genotype | Isolates | Identity (%) | ORF no. | Identity (%) | ORF no. | Identity (%) | ORF no. | Identity (%) | ORF no. | Identity (%) |
| ORF 056L | 59,881 | 60,672 | 792 | hypothetical protein | RSIV subtype II | RSIV KagYT-96 RSIV RIE12-1 GSIV-K1 LYCIV Zhoushan RBIV-KOR-TY1 OSGIV | 92.12% | ORF 055L | 95.58% | ORF 333L | 98.86% | 15R | 94.44% | 15R | 93.43% |
| ORF 057L | 60,678 | 61,652 | 975 | hypothetical protein | RSIV subtype II | RSIV KagYT-96 RSIV RIE12-1 GSIV-K1 LYCIV Zhoushan OSGIV | 100.00% | ORF 056L | 99.90% | ORF 342L | 97.03% | 14R | 92.31% | 14R | 92.23% |
| ORF 058L | 61,907 | 63,304 | 1398 | serine/threonine protein kinase | RSIV subtype II | RSIV KagYT-96 RSIV RIE12-1 OSGIV | 100.00% | ORF 057L | 99.93% | ORF 349L | 97.49% | 13R | 90.19% | 13R | 91.91% |
| ORF 059L | 63,311 | 63,643 | 333 | RING-finger-containing ubiquitin ligase | RSIV subtype II | RSIV KagYT-96 RSIV RIE12-1 GSIV-K1 RBIV-C1 LYCIV Zhoushan RSIV_121 RBIV-KOR-TY1 OSGIV 17RbGs | 100.00% | ORF 058L | 100.00% | ORF 350L | 98.50% | 12R | 96.36% | 12R | 95.80% |
| ORF 060R | 63,662 | 63,922 | 261 | hypothetical protein | RSIV subtype II | RSIV KagYT-96 RSIV RIE12-1 GSIV-K1 RBIV-C1 LYCIV Zhoushan RSIV_121 OSGIV | 100.00% | ORF 059R | 98.04% | ORF 351R | 96.55% | 11L | 95.02% | 11L | 94.90% |
| ORF 061R | 63,919 | 64,311 | 393 | hypothetical protein | RSIV subtype II | RSIV KagYT-96 RSIV RIE12-1 RBIV-C1 RSIV_121 RBIV-KOR-TY1 | 100.00% | ORF 060R | 92.11% | ORF 353R | 97.96% | 10L | 92.62% | 10L | 92.11% |
| ORF 062L | 64,470 | 64,631 | 162 | hypothetical protein | RSIV subtype II | RSIV KagYT-96 RSIV RIE12-1 GSIV-K1 RBIV-C1 LYCIV Zhoushan RSIV_121 RBIV-KOR-TY1 OSGIV 17RbGs | 100.00% | ORF 061L | 100.00% | ORF 360L | 98.77% | 9R | 97.53% | 9R | 98.77% |

**Table A2.** *Cont.*

| Gene ID 17SbTy | Position | | CDS Size (NT) | Predicted Structure and/or Function | Best-Match Homolog | | | Homolog to 17RbGs | | Homolog to Ehime_1 (AB104413.1) | | Homolog to ISKNV (AF371960) | | Homolog to TRBIV (GQ273492) | |
|---|---|---|---|---|---|---|---|---|---|---|---|---|---|---|---|
| | Start | End | | | Genotype | Isolates | Identity (%) | ORF no. | Identity (%) | ORF no. | Identity (%) | ORF no. | Identity (%) | ORF no. | Identity (%) |
| ORF 063L | 64,727 | 66,274 | 1548 | hypothetical protein | RSIV subtype II | RSIV KagYT-96 RSIV RIE12-1 GSIV-K1 RBIV-C1 LYCIV Zhoushan RSIV_121 OSGIV 17RbGs | 100.00% | ORF 062L | 100.00% | ORF 373L | 96.13% | 8R | 91.88% | 8R | 91.68% |
| ORF 064R | 66,345 | 67,802 | 1458 | myristoylated membrane protein | RSIV subtype II | RSIV KagYT-96RSIV RIE12-1GSIV-K1LYCIV ZhoushanOSGIV | 100.00% | ORF 063R | 99.73% | ORF 374R | 97.46% | 7L | 94.51% | 7L | 94.65% |
| ORF 065R | 67,819 | 69,180 | 1362 | major capsid protein | RSIV subtype I | LYCIV Zhoushan | 100.00% | ORF 064R | 98.24% | MCP | 99.63% | 6L | 94.57% | 6L | 94.27% |
| ORF 066R | 69,326 | 70,090 | 765 | NIF-NLI interacting factor-like phosphatase | RSIV subtype I | PIV2016 PIV2014a PIV2010 LYCIV Zhoushan RSIV Ehime-1 LYCIV | 100.00% | ORF 065R | 98.35% | ORF 385R | 100.00% | 5L | 95.17% | 5L | 92.82% |
| ORF 067R | 70,164 | 70,340 | 177 | hypothetical protein | RSIV subtype I | PIV2016 PIV2014a PIV2010 LYCIV Zhoushan RSIV Ehime-1 LYCIV | 100.00% | ORF 066R | 99.44% | ORF 388R | 100.00% | 4L | 91.78% | 4L | 97.89% |
| ORF 068R | 70,413 | 71,196 | 486 | hypothetical protein | RSIV subtype I | LYCIV Zhoushan | 100.00% | ORF 067R | 96.30% | ORF 390R | 99.79% | 3L | 90.00% | | 86.59% |
| ORF 069R | 71,268 | 71,735 | 468 | DNA dependent RNA polymerase subunit H like protein | RSIV subtype I | PIV2016 PIV2014a PIV2010 LYCIV Zhoushan RSIV Ehime-1 LYCIV | 100.00% | ORF 068R | 99.36% | RPOH | 100.00% | 2R | 93.83% | 2R | 94.25% |
| ORF 070R | 71,705 | 72,841 | 1137 | probable trans-membrane amino acid transporter | RSIV subtype I | PIV2016 PIV2014a PIV2010 LYCIV Zhoushan RSIV Ehime-1 LYCIV | 100.00% | ORF 069R | 97.89% | ORF 396R | 100.00% | 1L | 93.23% | 1L | 92.52% |

Table A2. *Cont.*

| Gene ID 17SbTy | Position | | CDS Size (NT) | Predicted Structure and/or Function | Best-Match Homolog | | Identity (%) | Homolog to 17RbGs | | Homolog to Ehime_1 (AB104413.1) | | Homolog to ISKNV (AF371960) | | Homolog to TRBIV (GQ273492) | |
| | Start | End | | | Genotype | Isolates | | ORF no. | Identity (%) | ORF no. | Identity (%) | ORF no. | Identity (%) | ORF no. | Identity (%) |
|---|---|---|---|---|---|---|---|---|---|---|---|---|---|---|---|
| ORF 071R | 72,956 | 73,672 | 717 | hypothetical protein | RSIV subtype II | RSIV RIE12-1 RSIV KagYT-96 GSIV-K1 OSGIV | 100.00% | ORF 070R | 99.86% | ORF 401R | 98.61% | 124L | 93.01% | 115L | 92.39% |
| ORF 072R | 73,681 | 74,061 | 381 | hypothetical protein | RSIV subtype II | RSIV KagYT-96 RSIV RIE12-1 GSIV-K1 RBIV-C1 RSIV_121 OSGIV 17RbGs | 100.00% | ORF 071R | 100.00% | ORF 407R | 98.69% | 123R | 97.58% | 114R | 95.90% |
| ORF 073L | 74,033 | 74,752 | 720 | ATPase(adenosine triphos-phatase) | RSIV subtype II | RSIV KagYT-96 RSIV RIE12-1 GSIV-K1 RBIV-C1 RSIV_121 OSGIV 17RbGs | 100.00% | ORF 072L | 100.00% | ORF 412L | 99.03% | 122R | 95.97% | 113R | 95.97% |
| ORF 074R | 74,762 | 75,397 | 636 | hypothetical protein | RSIV subtype II | RSIV KagYT-96 RSIV RIE12-1 GSIV-K1 RBIV-C1 RSIV_121 OSGIV | 97.48% | ORF 073R | 97.48% | ORF 413R | 97.16% | 121L | 86.09% | 112L | 84.54% |
| ORF 075L | 75,418 | 75,924 | 507 | hypothetical protein | RSIV subtype II | RSIV KagYT-96 GSIV-K1 RBIV-C1 RSIV_121 OSGIV 17RbGs | 100.00% | ORF 074L | 100.00% | ORF 420L | 97.24% | 120R | 93.53% | 111R | 92.28% |
| ORF 076L | 75,955 | 76,242 | 288 | probable tran-scriptional activator RING-finger domain-containing E3 protein | RSIV subtype II | RSIV KagYT-96 RSIV RIE12-1 GSIV-K1 RBIV-C1 RSIV_121 OSGIV 17RbGs | 100.00% | ORF 075L | 100.00% | ORF 423L | 98.96% | 119R | 93.71% | 110R | 92.01% |
| ORF 077R | 76,312 | 77,625 | 1314 | ankyrin repeat-containing protein | RSIV subtype II | RSIV KagYT-96 RSIV RIE12-1 GSIV-K1 | 100.00% | ORF 076R | 99.77% | ORF 424R | 96.88% | 118L | 93.03% | 109L | 92.03% |

**Table A2.** *Cont.*

| Gene ID 17SbTy | Position | | CDS Size (NT) | Predicted Structure and/or Function | Best-Match Homolog | | | Homolog to 17RbGs | | Homolog to Ehime_1 (AB104413.1) | | Homolog to ISKNV (AF371960) | | Homolog to TRBIV (GQ273492) | |
|---|---|---|---|---|---|---|---|---|---|---|---|---|---|---|---|
| | Start | End | | | Genotype | Isolates | Identity (%) | ORF no. | Identity (%) | ORF no. | Identity (%) | ORF no. | Identity (%) | ORF no. | Identity (%) |
| ORF 078R | 77,958 | 78,632 | 675 | FV3 early 31KDa protein homolog | RSIV subtype II | RSIV KagYT-96 GSIV-K1 RSIV_121 OSGIV | 99.85% | ORF 077R | 99.85% | ORF 436R | 98.22% | 117L | 93.79% | 108L | 94.82% |
| ORF 079L | 78,686 | 80,062 | 1377 | hypothetical protein | RSIV subtype II | RSIV KagYT-96 RSIV RIE12-1 GSIV-K1 17RbGs | 100.00% | ORF 078L | 100.00% | ORF 450L | 96.27% | 116R | 86.68% | 107R | 85.92% |
| ORF 080L | 80,123 | 81,133 | 1011 | immediate-early protein ICP46 | RSIV subtype II | RSIV KagYT-96 RSIV RIE12-1 GSIV-K1 RBIV-C1 RSIV_121 17RbGs | 100.00% | ORF 079L | 100.00% | ORF 458L | 98.32% | 115R | 93.18% | 106R | 93.08% |
| ORF 081R | 81,568 | 84,150 | 2583 | putative tyrosine kinase | RSIV subtype II | GSIV-K1 | 100.00% | ORF 080R | 99.96% | ORF 463R | 97.99% | 114L | 93.69% | 105L | 93.26% |
| ORF 082L | 84,194 | 84,574 | 381 | hypothetical protein | RSIV subtype II | RSIV KagYT-96 RSIV RIE12-1 GSIV-K1 RBIV-C1 RSIV_121 OSGIV | 100.00% | ORF 081L | 99.74% | ORF 483L | 97.38% | 113R | 92.66% | 104R | 92.89% |
| ORF 083L | 84,682 | 85,425 | 744 | proliferating cell nuclear antigen | RSIV subtype II | RSIV KagYT-96 RSIV RIE12-1 GSIV-K1 RBIV-C1 RSIV_121 OSGIV 17RbGs | 100.00% | ORF 082L | 100.00% | ORF 487L | 98.39% | 112R | 94.35% | 102R | 96.01% |
| ORF 084R | 85,445 | 86,341 | 897 | tumor necrosis factor recepter - assosiated factor-like protein | RSIV subtype II | RSIV KagYT-96RSIV RIE12-1GSIV-K1RBIV-C1RSIV_12117RbGs | 100.00% | ORF 083R | 100.00% | ORF 488R | 97.99% | 111L | 93.09% | 101L | 90.41% |
| ORF 085L | 86,338 | 86,493 | 156 | hypothetical protein | RSIV subtype II | RSIV KagYT-96 RSIV RIE12-1 GSIV-K1 RBIV-C1 RSIV_121 RBIV-KOR-TY1 OSGIV 17RbGs | 100.00% | ORF 084L | 100.00% | ORF 492L | 96.79% | 110R | 90.38% | 100R | 91.03% |

Table A2. *Cont.*

| Gene ID 17SbTy | Position | | CDS Size (NT) | Predicted Structure and/or Function | Best-Match Homolog | | | Homolog to 17RbGs | | Homolog to Ehime_1 (AB104413.1) | | Homolog to ISKNV (AF371960) | | Homolog to TRBIV (GQ273492) | |
|---|---|---|---|---|---|---|---|---|---|---|---|---|---|---|---|
| | Start | End | | | Genotype | Isolates | Identity (%) | ORF no. | Identity (%) | ORF no. | Identity (%) | ORF no. | Identity (%) | ORF no. | Identity (%) |
| ORF 086R | 86,546 | 89,308 | 2763 | D5 family NTPase | RSIV subtype II | RSIV KagYT-96 RSIV RIE12-1 GSIV-K1 OSGIV | 100.00% | ORF 085R | 99.96% | ORF 493R | 97.79% | 109L | 94.29% | 99L | 94.53% |
| ORF 087R | 89,389 | 90,018 | 630 | hypothetical protein | RSIV subtype II | RSIV RIE12-1 GSIV-K1 RBIV-C1 RSIV_121 RBIV-KOR-TY1 OSGIV | 99.84% | ORF 086R | 99.84% | ORF 502R | 95.67% | 108.5L | 91.61% | 98L | 94.91% |
| ORF 088R | 90,058 | 90,930 | 873 | hypothetical protein | RSIV subtype II | RSIV KagYT-96 RSIV RIE12-1 GSIV-K1 RBIV-C1 OSGIV 17RbGs | 100.00% | ORF 087R | 100.00% | ORF 506R | 97.25% | - | - | 97L | 80.25% |
| ORF 089L | 90,937 | 91,901 | 888 | HIT-like protein | RSIV subtype II | RSIV KagYT-96 RSIV RIE12-1 GSIV-K1 OSGIV | 99.89% | ORF 088L | 99.55% | ORF 515L | 96.83% | - | - | - | - |
| ORF 090L | 91,953 | 92,324 | 372 | hypothetical protein | RSIV subtype II | RSIV KagYT-96 RSIV RIE12-1 GSIV-K1 RBIV-C1 RSIV_121 RBIV-KOR-TY1 OSGIV 17RbGs | 100.00% | ORF 089L | 100.00% | ORF 518L | 98.66% | 105R | 95.99% | 96R | 94.62% |
| ORF 091L | 92,326 | 93,102 | 777 | hypothetical protein | RSIV subtype II | RSIV KagYT-96 RSIV RIE12-1 GSIV-K1 RBIV-C1 RSIV_121 OSGIV | 98.71% | ORF 090L | 98.71% | ORF 522L | 98.20% | 104R | 94.21% | 95R | 90.09% |
| ORF 092L | 93,164 | 93,577 | 414 | suppressor of cytokine signalling 1 homolog | RSIV subtype I | PIV2016 PIV2014a PIV2010 LYCIV Zhoushan RSIV Ehime-1 | 100.00% | ORF 091L | 95.17% | ORF 524L | 100.00% | 103R | 88.38% | 94R | 88.22% |
| ORF 093L | 93,584 | 95,029 | 1446 | ankyrin repeat containing protein | RSIV subtype I | PIV2016 PIV2014a PIV2010 LYCIV Zhoushan RSIV Ehime-1 | 100.00% | ORF 092L | 97.99% | ORF 534L | 100.00% | 102R | 91.46% | 93R | 92.39% |
| ORF 094R | 95,098 | 95,613 | 516 | hypothetical protein | RSIV subtype I | PIV2016 PIV2014a PIV2010 LYCIV Zhoushan RSIV Ehime-1 | 100.00% | ORF 093R | 97.29% | ORF 535R | 100.00% | 101L | 93.80% | 92L | 92.83% |

Table A2. *Cont.*

| Gene ID 17SbTy | Position | | CDS Size (NT) | Predicted Structure and/or Function | Best-Match Homolog | | Identity (%) | Homolog to 17RbGs | | Homolog to Ehime_1 (AB104413.1) | | Homolog to ISKNV (AF371960) | | Homolog to TRBIV (GQ273492) | |
|---|---|---|---|---|---|---|---|---|---|---|---|---|---|---|---|
| | Start | End | | | Genotype | Isolates | | ORF no. | Identity (%) | ORF no. | Identity (%) | ORF no. | Identity (%) | ORF no. | Identity (%) |
| ORF 095R | 95,588 | 96,229 | 642 | hypothetical protein | RSIV subtype II | RSIV KagYT-96 RSIV RIE12-1 GSIV-K1 OSGIV | 99.07% | ORF 094R | 98.75% | ORF 539R | 98.91% | 100L | 86.49% | 91L | 86.67% |
| ORF 096R | 96,283 | 96,606 | 324 | RING-finger-containing E3 ubiquitin ligase | RSIV subtype II | RSIV KagYT-96 GSIV-K1 RBIV-C1 RSIV_121 RBIV-KOR-TY1 OSGIV 17RbGs | 100.00% | ORF 095R | 100.00% | ORF 543R | 97.53% | 99L | 91.05% | 90L | 84.26% |
| ORF 097R | 96,655 | 97,146 | 492 | hypothetical protein | RSIV subtype II | RSIV KagYT-96 RSIV RIE12-1 GSIV-K1 RBIV-C1 RSIV_121 OSGIV 17RbGs | 100.00% | ORF 096R | 100.00% | ORF 545R | 97.36% | 97.5L | 94.51% | 89L | 92.48% |
| ORF 098R | 97,137 | 97,888 | 738 | hypothetical protein | RSIV subtype II | RSIV KagYT-96 RSIV RIE12-1 GSIV-K1 RBIV-C1 RSIV_121 OSGIV 17RbGs | 100.00% | ORF 097R | 100.00% | ORF 550R | 98.10% | 96L | 94.58% | 88L | 93.77% |
| ORF 099R | 97,896 | 99,059 | 1164 | hypothetical protein | RSIV subtype II | RSIV KagYT-96 RSIV RIE12-1 GSIV-K1 OSGIV | 100.00% | ORF 098R | 99.91% | ORF 554R | 96.91% | 95L | 91.21% | 87L | 91.02% |
| ORF 100R | 99,084 | 99,584 | 501 | hypothetical protein | RSIV subtype II | RSIV KagYT-96 RSIV RIE12-1 GSIV-K1 RBIV-C1 RSIV_121 OSGIV 17RbGs | 100.00% | ORF 099R | 100.00% | ORF 562R | 98.60% | 94L | 95.41% | 86L | 93.01% |
| ORF 101R | 99,594 | 100,520 | 927 | probable RNA binding protein | RSIV subtype II | RSIV KagYT-96RSIV RIE12-1GSIV-K1SKIVRBIV-C1RSIV_121OSGIV17RbGs | 100.00% | ORF 100R | 100.00% | ORF 569R | 97.84% | 93L | 92.22% | 85L | 92.02% |
| ORF 102R | 100,641 | 101,711 | 1071 | myristoylated membrane protein | RSIV subtype II | RSIV KagYT-96 RSIV RIE12-1 GSIV-K1 OSGIV | 98.62% | ORF 101R | 99.69% | ORF 575R | 95.94% | - | - | 83L | 91.36% |
| ORF 103L | 101,692 | 103,263 | 1572 | hypothetical protein | RSIV subtype I | PIV2016 PIV2014a PIV2010 LYCIV Zhoushan | 98.85% | ORF 102L | 98.54% | ORF 586L | 98.20% | 88R | 92.24% | 82R | 93.26% |

Table A2. *Cont.*

| Gene ID 17SbTy | Position | | CDS Size (NT) | Predicted Structure and/or Function | Best-Match Homolog | | Identity (%) | Homolog to 17RbGs | | Homolog to Ehime_1 (AB104413.1) | | Homolog to ISKNV (AF371960) | | Homolog to TRBIV (GQ273492) | |
|---|---|---|---|---|---|---|---|---|---|---|---|---|---|---|---|
| | Start | End | | | Genotype | Isolates | | ORF no. | Identity (%) | ORF no. | Identity (%) | ORF no. | Identity (%) | ORF no. | Identity (%) |
| ORF 104R | 103,311 | 103,724 | 414 | hypothetical protein | RSIV subtype II | RSIV KagYT-96 RSIV RIE12-1 GSIV-K1 RBIV-C1 RSIV_121 OSGIV | 99.52% | ORF 103R | 99.52% | ORF 591R | 99.28% | | 94.31% | 81R | 95.48% |
| ORF 105L | 103,721 | 104,518 | 798 | RNase III-like ribonuclease | RSIV subtype II | RSIV KagYT-96 RSIV RIE12-1 RBIV-C1 RSIV_121 17RbGs | 100.00% | ORF 104L | 100.00% | RNC | 97.99% | 87R | 94.16% | 80R | 94.86% |
| ORF 106L | 104,484 | 104,951 | 468 | Uvr/REP helicase | RSIV subtype II | RSIV KagYT-96 RSIV RIE12-1 GSIV-K1 OSGIV | 100.00% | ORF 105L | 93.80% | ORF 600L | 97.44% | 86R | 92.55% | 79R | 92.95% |
| ORF 107L | 104,948 | 105,451 | 504 | hypothetical protein | RSIV subtype II | RSIV KagYT-96 RSIV RIE12-1 GSIV-K1 RBIV-C1 RSIV_121 RBIV-KOR-TY1 OSGIV | 100.00% | ORF 106L | 92.86% | ORF 605L | 97.83% | 85R | 92.74% | 78R | 90.73% |
| ORF 108R | 105,565 | 106,869 | 1305 | hypothetical protein | RSIV subtype II | RSIV KagYT-96 RSIV RIE12-1 GSIV-K1 OSGIV | 100.00% | ORF 107R | 95.21% | ORF 606R | 97.70% | 84L | 93.16% | 77L | 91.58% |
| ORF 109L | 106,896 | 107,255 | 360 | hypothetical protein | RSIV subtype II | RSIV KagYT-96 RSIV RIE12-1 GSIV-K1 RBIV-C1 RSIV_121 OSGIV | 100.00% | ORF 108L | 98.89% | ORF 617L | 98.33% | 83R | 93.46% | 76R | 92.48% |
| ORF 110R | 107,319 | 10,8425 | 1107 | hypothetical protein | RSIV subtype II | RSIV KagYT-96 RSIV RIE12-1 GSIV-K1 OSGIV | 100.00% | ORF 109R | 99.82% | ORF 618R | 97.92% | 82L | 93.59% | 75L | 93.22% |
| ORF 111L | 108,474 | 108,971 | 498 | hypothetical protein | RSIV subtype II | RSIV KagYT-96 RSIV RIE12-1 GSIV-K1 RBIV-C1 RSIV_121 OSGIV 17RbGs | 100.00% | ORF 110L | 100.00% | ORF 628L | 97.99% | 81R | 95.78% | 74R | 94.32% |
| ORF 112L | 108,984 | 109,457 | 474 | hypothetical protein | RSIV subtype II | RSIV KagYT-96 RSIV RIE12-1 GSIV-K1 RBIV-C1 RBIV-KOR-TY1 OSGIV 17RbGs | 100.00% | ORF 111L | 100.00% | ORF 632L | 95.81% | - | - | 73R | 85.56% |

**Table A2.** *Cont.*

| Gene ID 17SbTy | Position | | CDS Size (NT) | Predicted Structure and/or Function | Best-Match Homolog | | | Homolog to 17RbGs | | Homolog to Ehime_1 (AB104413.1) | | Homolog to ISKNV (AF371960) | | Homolog to TRBIV (GQ273492) | |
|---|---|---|---|---|---|---|---|---|---|---|---|---|---|---|---|
| | Start | End | | | Genotype | Isolates | Identity (%) | ORF no. | Identity (%) | ORF no. | Identity (%) | ORF no. | Identity (%) | ORF no. | Identity (%) |
| ORF 113R | 109,545 | 109,769 | 225 | hypothetical protein | RSIV subtype II | RSIV KagYT-96 RSIV RIE12-1 GSIV-K1 RBIV-C1 RSIV_121 OSGIV 17RbGs | 100.00% | ORF 112L | 100.00% | ORF 634L | 92.06% | 79L | 93.78% | 72L | 92.27% |
| ORF 114L | 109,771 | 110,235 | 465 | hypothetical protein | RSIV subtype II | RSIV KagYT-96 RSIV RIE12-1 GSIV-K1 RBIV-C1 RSIV_121 OSGIV 17RbGs | 100.00% | ORF 113L | 100.00% | ORF 635L | 97.42% | 78R | 96.34% | 71R | 93.76% |
| ORF 115L | 110,232 | 111,566 | 1335 | hypothetical protein | RSIV subtype II | RSIV KagYT-96 RSIV RIE12-1 GSIV-K1 RBIV-C1 OSGIV | 99.93% | ORF 114L | 99.93% | ORF 641L | 96.55% | 77R | 90.95% | 70R | 90.42% |

**Table A3.** The coding sequences (CDSs) determined via COG classification of 17SbTy and 17RbGs in four functional categories.

| No. | Category | COG Function | COG Descrption | 17SbTy | 17RbGS |
|---|---|---|---|---|---|
| 1 | Metabolism | Amino acid transport metabolism | quinoprotein dehydrogenase-associated putative ABC transporter substrate-binding protein | ORF 093L | ORF 092L |
| 2 | | Nucleotide transport and metabolism | deoxynucleoside kinase | ORF 042L | ORF 041L |
| 3 | | | ribonucleoside-diphosphate reductase | ORF 048L | ORF 047L |
| 4 | | | HIT domain-containing protein | ORF 089L | ORF 088L |
| 5 | Information storage and processing | Translation, ribosomal structure and biogenesis | O-acetyl-ADP-ribose deacetylase | ORF 050R | ORF 049R |
| 6 | | Transcription | DNA-directed RNA polymerase subunit B | ORF 040L | ORF 039L |
| 7 | | | transcription factor S | ORF 044R | ORF 043R |
| 8 | | | DNA-directed RNA polymerase subunit A&apos | ORF 045R | ORF 044R |
| 9 | | | phosphoprotein phosphatase | ORF 066R | ORF 065R |
| 10 | | | ribonuclease III | ORF 105L | ORF 104L |
| 11 | | Replication, recombination and repair | DNA cytosine methyltransferase | ORF 028R | ORF 027R |
| 12 | | | flap endonuclease-1 | ORF 046R | ORF 045R |
| 13 | | | DNA polymerase elongation subunit | ORF 052L | ORF 051L |
| 14 | Cellular process | Signal transduction mechanisms | protein-tyrosine-phosphatase | ORF 12R | ORF 012R |
| 15 | | | ankyrin repeat-containing protein | ORF 077R | ORF 076R |
| 16 | | | quinoprotein dehydrogenase-associated putative ABC transporter substrate-binding protein | ORF 093L | ORF 092L |
| 17 | | | ankyrin repeat-containing protein | ORF 115L | ORF 114L |
| 18 | | Mobilome; prophages, transposons | hypothetical protein | ORF 086R | ORF 085R |
| 19 | Poorly characterized | General function prediction only | HIT domain-containing protein | ORF 089L | ORF 088L |
| 20 | | Function unknown | hypothetical protein | ORF 013R | ORF 012R |

**Table A4.** ORF locations of the 26 conserved core genes conserved in the family *Iridoviridae*.

| No. | Gene (GenBank Access. No.) | 17SbTy (OK042108) | 17RbGs (OK042109) | Ehime-1 (AB104413) | ISKNV (AF371960) | RBIV (AY532606) | TRBIV (GQ273492) |
|---|---|---|---|---|---|---|---|
| 1 | hypothetical protein | 001R | 001R | 639R | 76L | 72L | 69L |
| 2 | Putative NTPase I | 013R | 012R | NTPase | 63L | 59L | 58L |
| 3 | Putative replication factor and/or DNA binding-packing | 015R | 014R | 092R | 61L | 57L | 56L |
| 4 | Helicase family | 018R | 017R | 101R | 56L | 54L | 53L |
| 5 | Serine-threonine protein kinase | 019R | 018R | 106R | 55L | 53L | 52L |
| 6 | Erv1/Alr family | 031R | 030R | 156R | 43L | 43.5L | 42L |
| 7 | DNA dependent RNA polymerase second largest subunit | 040L | 039L | RPO-2 | 34R | 33R | 33R |
| 8 | Deoxynucleoside kinase | 042L | 041L | TK | 32R | 31R | 31R |
| 9 | Transcription elongation factor TFIIS | 044R | 043R | 238R | 29L | 29.5Lb | 29L |
| 10 | DNA dependent RNA polymerase largest subunit | 045R | 044R | RPO-1 | 28L | 29L | 26L |
| 11 | Putative XPPG-RAD2-type nuclease | 046R | 045R | 256R | 27L | 28L | 27L |
| 12 | Ribonucleotide reductase small subunit | 048L | 047L | RR-2 | 24R | 26R | 25R |
| 13 | DNA pol Family B exonuclease | 052L | 051L | DPO | 19R | 20R | 20R |
| 14 | Serine-threonine protein kinase | 058L | 057L | 349L | 13R | 13R | 13R |
| 15 | Myristoylated membrane protein | 064R | 063R | 374R | 7L | 8L | 7L |
| 16 | Major capsid protein | 065R | 064R | MCP | 6L | 7L | 6L |
| 17 | NIF-NLI interacting factor | 066R | 065R | 385R | 5L | 6L | 5L |
| 18 | ATPase(adenosine triphosphatase) | 073L | 072L | 412L | 122R | 116R | 113R |
| 19 | Immediate early protein ICP-46 | 080L | 079L | 458L | 115R | 108.5R | 106R |
| 20 | Putative tyrosin kinase/lipopolysaccharide modifying enzyme | 081R | 080R | 463R | 61L, 114L | 57L, 106Lb | 105L |
| 21 | Proliferating cell nuclear antigen | 083L | 082L | 487L | 112R | 103Rb | 102R |
| 22 | D5 family NTPase involved in DNA replication | 086R | 085R | 493R | 109L | 101L | 99L |
| 23 | hypothetical protein | 098R | 097R | 550R | 96L | 89.5Lb | 88L |
| 24 | Myristoylated membrane protein | 102R | 101R | 575R | 90.5L | 85L | 83R |
| 25 | RNase III-like ribonuclease | 105L | 104L | RNC | 87R | 83R | 80R |
| 26 | Uvr/REP helicase | 106L | 105L | 600L | 86R | 82.5R | 79R |

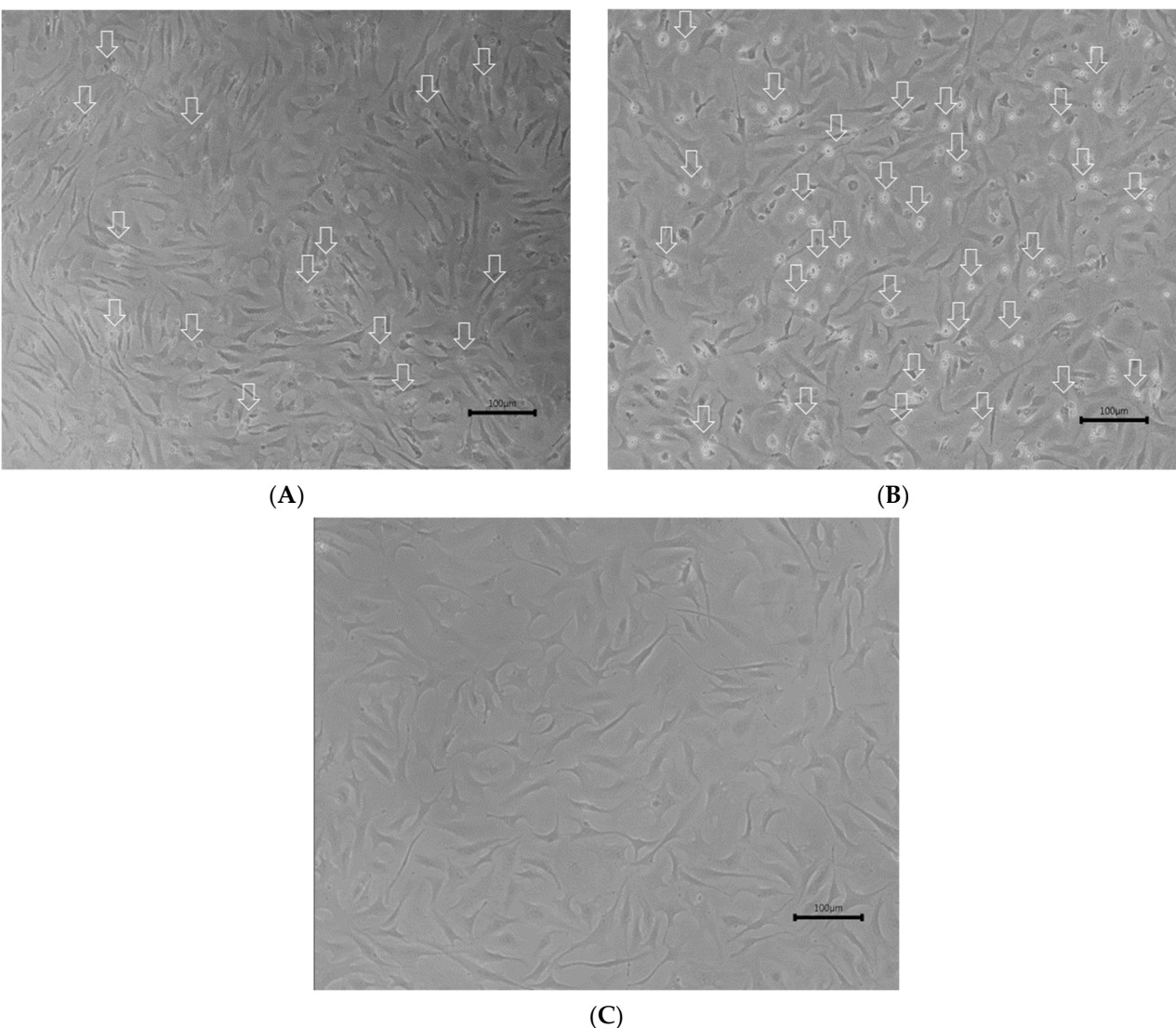

(A)

(B)

(C)

**Figure A1.** Cytopathic effects (CPEs) in rock bream fin cells under the influence of a tissue homogenate from (**A**) an RSIV (17SbTy)-infected Japanese seabass and (**B**) an RSIV (17RbGs)-infected rock bream. CPE of the rounding cells (arrows) in rock bream fin cells (**A**) after 3 days of inoculation with 17SbTy, and (**B**) 9 days of inoculation with 17RbGs, and (**C**) negative control (mock cells at passage 15). Scale bar = 100 μm.

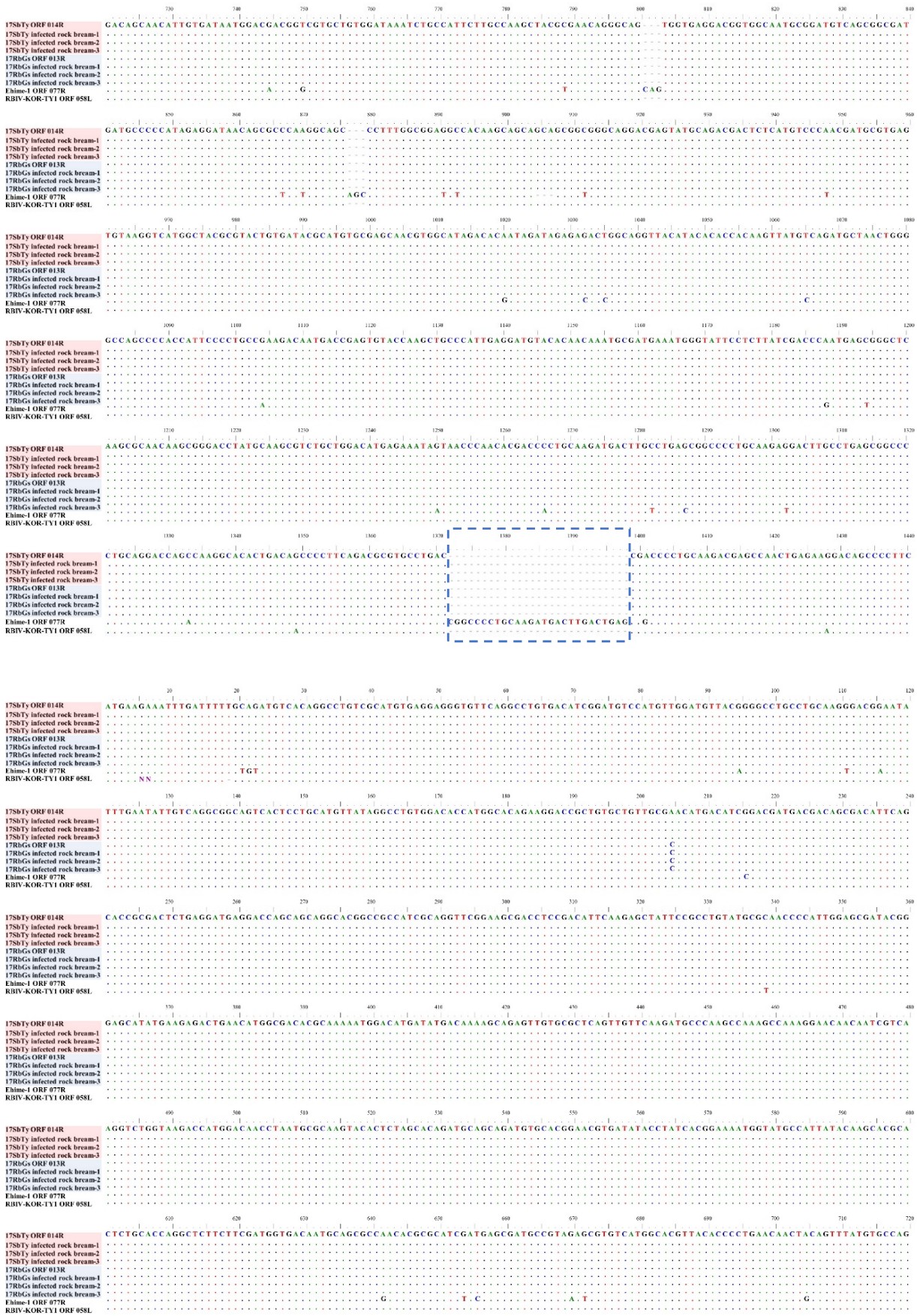

**Figure A2.** *Cont.*

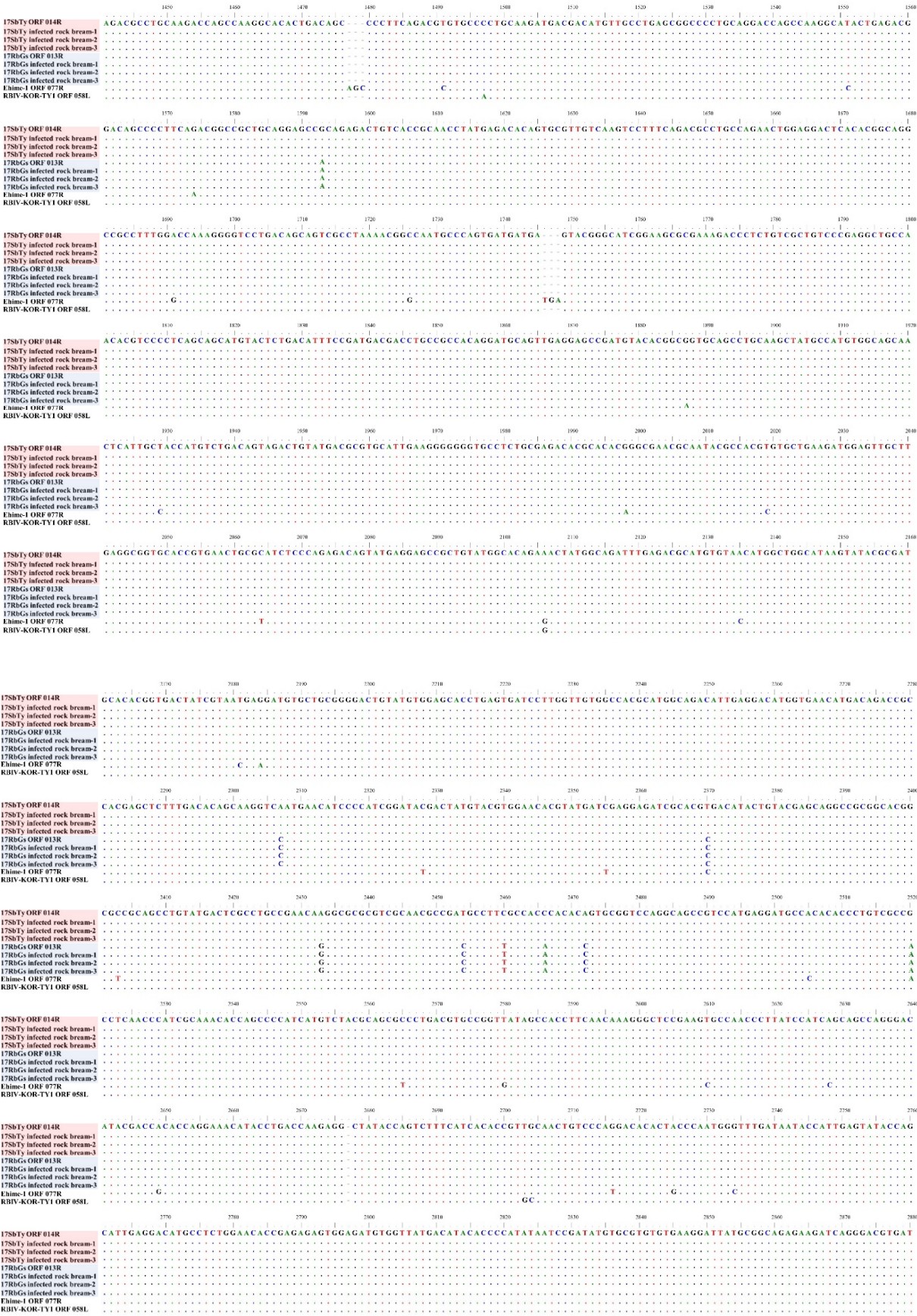

**Figure A2.** *Cont.*

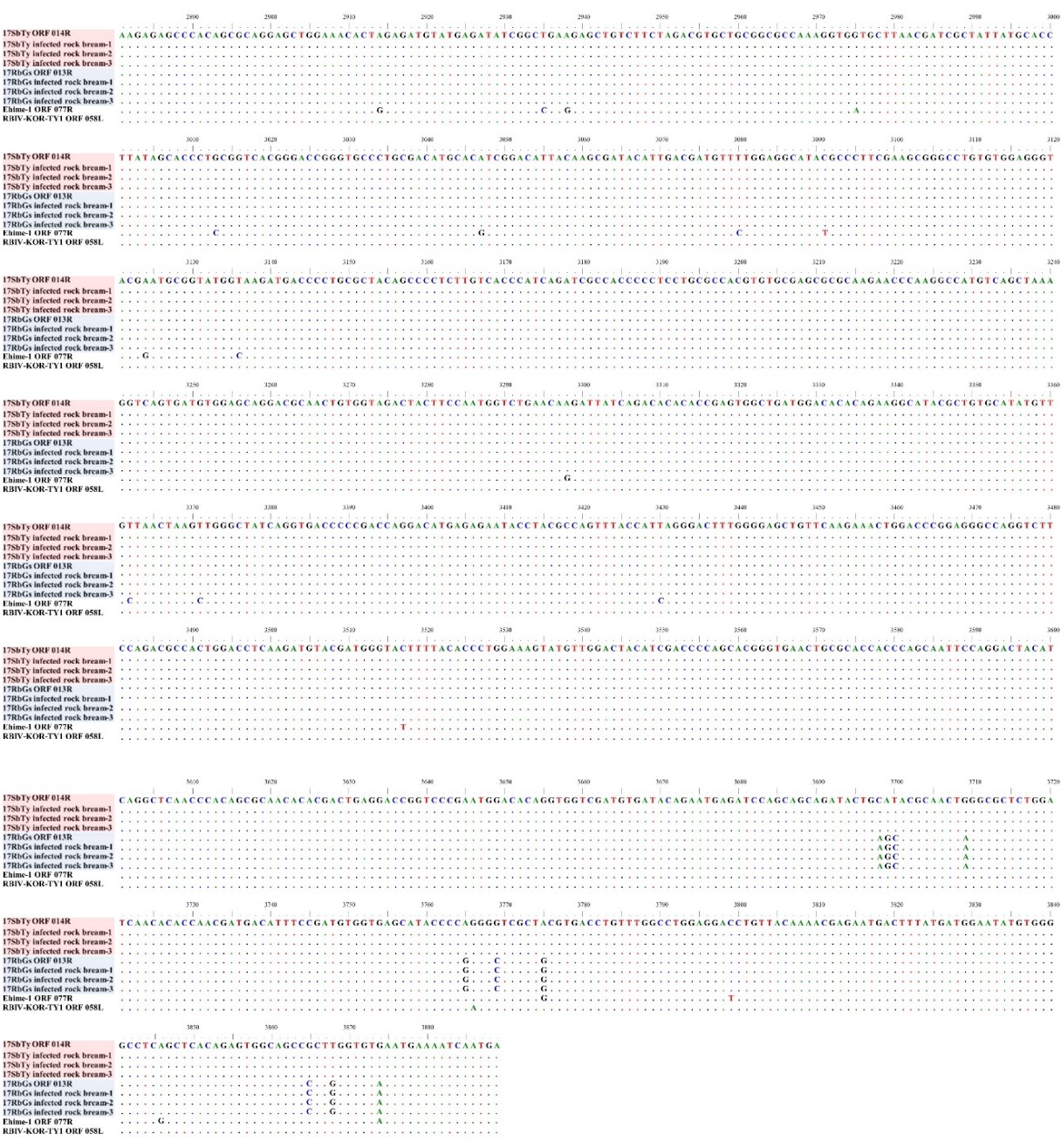

(**a**)

**Figure A2.** *Cont.*

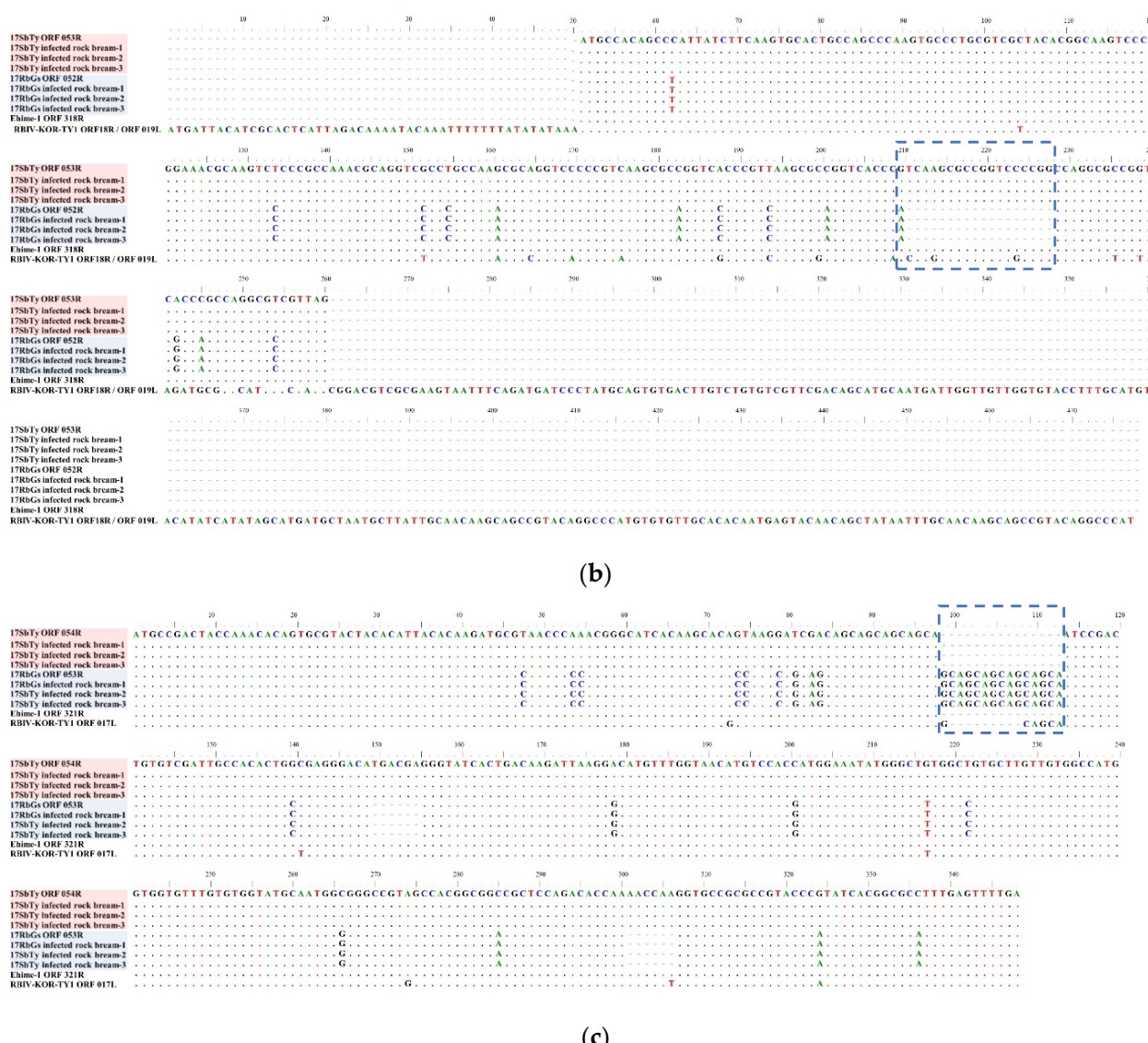

(**b**)

(**c**)

**Figure A2.** *Cont.*

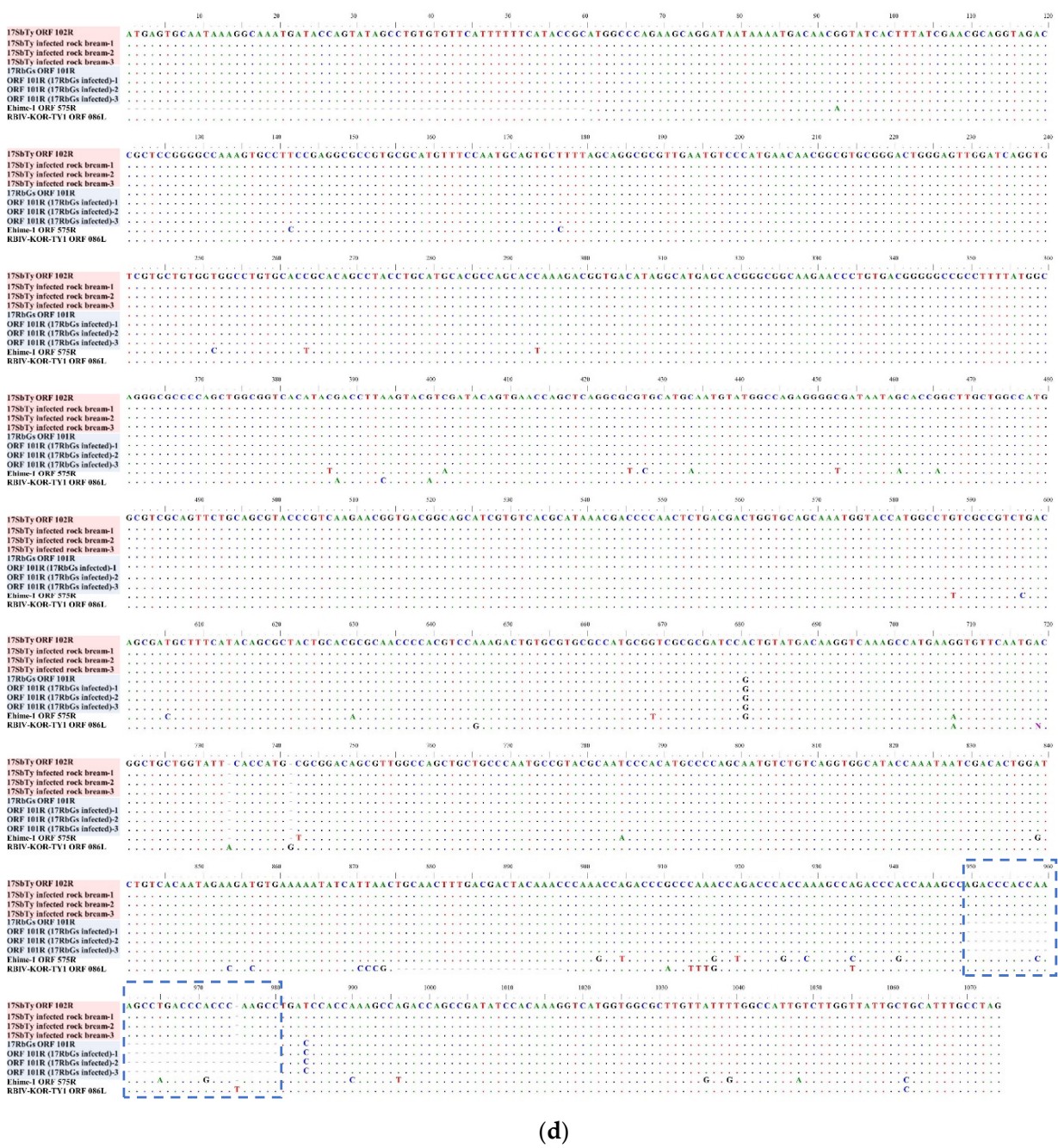

(**d**)

**Figure A2.** Comparison of nucleotide sequences covering the four insertion and deletions (InDels) in coding regions (ORFs (**a**) 014R, (**b**) 053R, (**c**) 054R and (**d**) 102R on the basis of 17SbTy isolate) between the cell-cultured isolates and viruses from RSIV-infected rock breams. The 17SbTy and 17RbGs from either cell-isolates or viruses from RSIV-infected rock bream are highlighted in red and blue boxes, respectively. The boxes consisting of blue dashed lines represent the InDel regions.

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
