# Peer review of "Complete Genome Sequences and Pathogenicity Analysis of Two Red Sea Bream Iridoviruses Isolated from Cultured Fish in Korea"

_fishes, doi:10.3390/fishes6040082_

Round 1

Reviewer 1 Report

I found this report interesting in that it identifies a possible natural recombinant of RSIV subtype I and II that has increased pathogenicity for rock bream.  It does raise questions regarding the role of other viral genes involved in virulence.

Author Response

Response: We agree with your comments. In this study, we focused on the determination of complete genome sequences of RSIVs showing different subtype patterns (17SbTy as RSIV mixed subtype I/II; 17RbGs as RSIV subtype II). We identified four InDels in coding regions (ORFs 014R, 053R, 054R, and 102R on the basis of the 17SbTy isolate). Of these InDels, two affect sequences of functional proteins (ORF014R for a DNA-binding protein and ORF102R for a myristoylated membrane protein) associated with viral replication and/or pathogenicity in the hosts. Furthermore, we found a significant difference in survival rates between bream groups after the challenge test of the RSIV isolates. As per your comments, we are trying to determine the functional role of other viral genes involved in virulence by assaying the expression levels of viral genes and immune genes and via transcriptome analysis post-infection. However, as you are aware, it is difficult to identify the functional roles of genes without a variety of experiments because the factors associated with virulence (viral replication, host immune system, etc.) are numerous and diverse. We appreciate your constructive comments; however, substantial time is required to identify the correlations between viral processes and the host immune system. Thus, within the scope of this manuscript, it is difficult to provide a definite answer regarding other viral genes involved in virulence. Please consider the time limits for this revision and the considerable time needed for several experiments. We will address your suggestion in further studies.

Reviewer 2 Report

In this work, two complete genomes of fish iridoviruses are presented and compared one to another and to other related strains. The two viruses are globally similar one to another and two others published. Authors focus on a few indels differing between their two viruses and two others. Finally, an infectious challenge compares the virulence of both viruses on rock-bream.

Because the paper focuses mainly on the molecular biology of viruses, its place in a journal like FISH should be reconsidered, despite the fact that FISH can accept papers related to diseases. Why not targeting a more specialized journal ?

I would recommend acceptation but after major corrections:

A major issue is the description of mutations between viruses that have been cell cultured and the proposed hypothesis regarding the impact of these mutations on the pathogenicity in fish. It would have been of high interest to verify the presence of these indels in the original infected fish (and why not after experimental infection), by a simple PCR/sequencing. If this work can not be done, the authors must mention the possibility that these mutations were produced during the cell culture.

Another point is that the apparent differences of virulence between the viruses may be due to adaptations to their respective original hosts. This must be discussed.

The text (including the abstract !) must absolutely be reviewed by an expert proficient in English. Some sentences are difficult to understand, too long or with poor grammar (for instance, line 52, or lines 59-61 or 78-80). Some words are missing (for instance lines 19 and line 93).

Part 2.4.3.

What about substitutions ?

Line 185-186. Rephrase. For instance: “ 17SbTY is grouped with subtypes I or subtypes II of RSIV depending on the considered gene, MCP or ATPase, respectively.”

Lines 191-192 are a repeat of 185-186 (same idea).

Line 195-196. Rephrase.

Line 262. What is the impact of the deletion on the size of ORF012R ? Is this deletion also present in the fish, and not only in the cultured isolate ?

Figure 4. Should position 100168 of virus RBIV (ORF86L) be 100138 instead ?

Author Response

We have made every effort to improve the quality of the revised manuscript according to the reviewers’ suggestions. In accordance with your suggestions, we have had the manuscript fully checked and revised by native speakers to improve grammar and language. Besides, we have performed the additional experiments comparing the nucleotide sequences of four InDels (ORFs 014R, 053R, 054R, and 102R on the basis of 17SbTy isolate) between cell-cultured isolates and viruses from RSIV-infected rock breams. We found that the InDel sequences perfectly match between the two groups. Furthermore, in the revised manuscript, we have added another possible explanation for the virulence difference between the two RSIV isolates: adaptations to their original hosts. Pleased find detail response in the attached file.

Reviewer 3 Report

The authors described a comparative genomics paper focusing on two gene coding DNA binding protein and myristylated membrane protein (ORF014R and ORF102R) in regions encoding functional proteins having insertion and deletion mutations.

Authors detected the survival rates of RSIV infected‒ rock bream differed significantly from those of the uninfected ones, indicating that the genomic characteristics might influence pathogenicity.

The paper is focused on important genomic features to unravel pathogenicity related genes and their impact on fish survival.

The complete genomes are well described for 17SbTy (RSIV subtype I/II) from Japanese seabass (Lateolabrax japonicas) and 17RbGs (RSIV subtype II) from rock bream (Oplegnathus fasciatus). However, the authors did not choose to use either iridoviral core genes or complete genome information for the evaluation of both phylogenetic analysis and functional comparisons. The complete genome-based analysis methodology has already been described Ince et al. 2018,  https://doi.org/10.3390/v10040161

-Including the images from primary cell culture monolayers and the infected cells showing CPE will be very important.

.-Figure 5 has an unclear 17RgBs annotation form.

Author Response

We have made every effort to improve the quality of the revised manuscript according to the reviewers’ suggestions. In accordance with reviewer your, we have additionally analyzed the phylogeny of 26 core genes that are shared by all members of the family Iridoviridae and have inserted a phylogenetic tree based on the concatenated amino acid sequences of the 26 conserved genes (Figure 3 in the revised manuscript). Moreover, we have inserted an image of the infected cells showing cytopathic effects as Figure A1. Below, please find our point-by-point responses to the referees’ comments. Please find the attached file.

Round 2

Reviewer 2 Report

The paper can be published. However, I still think that it would be better in a virological journal.

I have suggested this idea to the authors who pretended then that the editor of 'Pathogens' recommended them to submit it to 'Fish' !

Author Response

Thank you for your comments. Even we focused on the complete genome sequence of red sea bream iridovirus and its pathogenicity in this study, our research (Fish disease) is undeniable that it fit the scope of Fishes journal. In addition, since many valid revisions have been made by peer-review in the Fishes, coauthors don’t think it’s desirable to publish in another journal. Co-authors hope that the revised manuscript to be published in the Fishes

Reviewer 3 Report

The first round of comments was addressed very well and the manuscript has been improved significantly.

Why phylogenetic rooted without using outgroup? Some bootstrapping values are low and phylogenetic resolution is low.

Author Response

We agree with your comments. As suggested, we have reconstructed the phylogenetic trees (Figure 1a, b) with outgroup (i.e., epizootic hematopoietic necrosis virus). The genotypes (and RSIV subtypes) were supported by bootstrap values >70%, and low bootstrap values (i.e., <70%) are not shown in the figure. The original high-resolution figures have been uploaded to the journal submission system.

This manuscript is a resubmission of an earlier submission. The following is a list of the peer review reports and author responses from that submission.

Round 1

Reviewer 1 Report

The paper describes the complete sequences of two genomes of megalocytivirus isolated in a previous work. Apart from this technical goal, the objectives of the paper are not clear. How does the “genome wide comparison” can play a “pivotal role” in an “advanced control strategy” ? by molecular tracing ? vaccine designing ? other ? Other issue: why choosing these two particular viruses for sequencing, especially 17RbGs for which features are not presented in the introduction (before line 51) ?

Since there are only sequences, very similar to others already published and obtained starting from viruses isolated in a previous paper, the information is of limited interest for the journal ‘Pathogens’. These results should be presented in another journal, more appropriate for new sequences of known viruses.

Other issues are:

- Because in vitro propagation can lead to mutations, it would have been of interest to check that some of the described mutations (SNP, indels), with a putative role in the biology of the virus, are also present in the fish.

- To complete the classification, the species ISKNV should be mentioned in the introduction

- L. 40. Indicate the host of RSIV-Ku isolated.

- L. 42. Is 17SbTy the same than SB5-TY reported in Kim 2019 ? it is confusing.

- L. 43-44. rephrase.

- L. 59. I don’t understand the term Uniform. Do you mean ‘reference’ ?

- L. 75. The Genbank accession numbers are not available (same at the end in the DAS).

- Line 219: unclear sentence; reformulate

- figure 1. Indicate in the legend the number of nt positions aligned (provided in the MEGA tree file).

- Figure 4: replace intragenic by intergenic

- L. 69. Finding a new variant on a single fish does not necessarily indicate that it is emerging. It could have been also just not found before if the survey was too small (time scale or geographical sites). This hypothesis should be considered too.

- L. 175-180. rephrase

Reviewer 2 Report

The present manuscript reports the complete genomes of 2 viruses of the species ISKNV, genus megalocytivirus from Korea. This is a very important pathogen in aquaculture and this additional study builds in a previous report about the pathogen in Korea.

Overall, complete genomes are important for understanding the emergence and dispersal of pathogens within the megalcytivirus genus and more specifically the ISKNV species. The aim of this study was to obtain more data describing the genetic variation of 2 apparently dissimilar variants of RSIV in Korea. More could be done to explain the origin of these samples and the reason they were selected for analysis. The genomes are not yet available to the public and more can be done to describe them e.g. the choice of analyses is not comprehensive.

There are specific comments in the attached PDF
